**Investigation**

# A new set of mutations in the second transmembrane helix of the Cox2p-W56R substantially improves its allotopic expression in *Saccharomyces cerevisiae*

Kewin Gombeau [ID] ,[1,2,]* Stefan A. Hoffmann [ID] ,[1,3] Yizhi Cai [ID] [1,2,]*

[1]Manchester Institute of Biotechnology, The University of Manchester, Manchester M1 7DN, UK
[2]Generative and Synthetic Genomics, Wellcome Sanger Institute, Cambridge CB10 1SA, UK
[3]Systems and Synthetic Biology, Wageningen University & Research, Wageningen 6708 WE, Netherlands

*Corresponding authors: Yizhi Cai, Department of Chemistry, Manchester Institute of Biotechnology, 131 Princess Street, Manchester M1 7DN, UK, Email: yizhi.cai@manchester.ac.uk; Kewin Gombeau, Department of Chemistry, Manchester Institute of Biotechnology, 131 Princess Street, Manchester M1 7DN, UK, Email: kewin.gombeau@manchester.ac.uk

The dual genetic control of mitochondrial respiratory function, combined with the high mutation rate of the mitochondrial genome (mtDNA), makes mitochondrial diseases among the most frequent genetic diseases in humans (1 in 5,000 in adults). With no effective treatments available, gene therapy approaches have been proposed. Notably, several studies have demonstrated the potential for nuclear expression of a healthy copy of a dysfunctional mitochondrial gene, referred to as allotopic expression, to help recover respiratory function. However, allotopic expression conditions require significant optimization. We harnessed engineering biology tools to improve the allotopic expression of the *COX2-W56R* gene in the budding yeast *Saccharomyces cerevisiae*. Through conducting random mutagenesis and screening of the impact of vector copy number, promoter, and mitochondrial targeting sequence, we substantially increased the mitochondrial incorporation of the allotopic protein and significantly increased recovery of mitochondrial respiration. Moreover, CN-PAGE analyses revealed that our optimized allotopic protein does not impact cytochrome c oxidase assembly, or the biogenesis of respiratory chain supercomplexes. Importantly, the most beneficial amino acid substitutions found in the second transmembrane helix (L93S and I102K) are conserved residues in the corresponding positions of human MT-CO2 (L73 and L75), and we propose that mirroring these changes could potentially help improve allotopic Cox2p expression in human cells. To conclude, this study demonstrates the effectiveness of using engineering biology approaches to optimise allotopic expression of mitochondrial genes in the baker's yeast.

Keywords: allotopic expression; engineering biology; yeast; mitochondria; random mutagenesis

## Introduction

Mitochondria are crucial organelles, producing the bulk of cellular energy through a process referred to as oxidative phosphorylation (Malina *et al.* 2018). Interestingly, during evolution, they retained a small, bacterial-like genome encoding only a few genes supporting energy production (Wallace 2007). This additional genome is a puzzling heritage of evolution, considering both the cost of its maintenance (Kelly 2021) and the risks of storing DNA in an organelle producing reactive oxygen species (Allen 2003; Wallace 2007). The high mutation rate of the mitochondrial genome (mtDNA), combined with dual genetic (mitochondrial and nuclear) control of mitochondrial function, makes mitochondrial diseases among the most frequent genetic diseases in humans (1 in 5,000 adults; Gorman *et al.* 2015). While no effective treatment has been discovered, gene therapy approaches offer a promising method to alleviate the deleterious effects of these mutations (Gorman *et al.* 2015). Among available gene therapy strategies, allotopic expression holds great potential but remains far from trivial to implement (Perales-Clemente *et al.* 2011; Soldatov *et al.* 2022).

This latter process involves the restoration of mitochondrial function through artificial nuclear relocation and expression of a healthy copy of an impaired mitochondrial gene. To this end, the relocated gene must be recoded to match the nuclear genetic code and associated with a mitochondrial targeting sequence (MTS) to direct the protein to its mitochondrial sub-compartment after cytoplasmic synthesis. This technique has been widely used in the past decades to understand the forces and constraints shaping mitochondrial DNA during evolution (Artika 2020). Recently, allotopic expression has been used to treat patients with Leber's hereditary optic neuropathy (Guo *et al.* 2015; Wan *et al.* 2016; Finsterer and Zarrouk-Mahjoub 2018; Newman *et al.* 2021). However, nuclear relocation does not guarantee proper mitochondrial import and sorting of the allotopically expressed protein, particularly due to the physico-chemical properties of the relocated proteins (Nieto-Panqueva *et al.* 2023). Indeed, as most mtDNA-encoded proteins are transmembrane proteins, their hydrophobicity can impair their mitochondrial import and sorting after cytoplasmic synthesis, even triggering mistargeting to other cellular compartments (Björkholm *et al.* 2015). In addition, mitochondrial proteins expressed from nuclear DNA require chaperones to remain soluble in the cytoplasm (Fan *et al.* 2006).

While error-prone PCR (Supekova *et al.* 2010) and protein fusions (Bietenhader *et al.* 2012) have been used to balance issues

**Table 1.** List of the strains used in this study.

| Strains IDs | Strains aliases | Nuclear genotype | Mitochondrial genotype | Source |
|---|---|---|---|---|
| YCy4097 | NB64 | MATa lys2 leu2-3,112 ura3-52 his3ΔHinDIII arg8::hisG | cox2-22 ρ$^+$ | Bonnefoy et al. 2001 |
| YCy3894 | DFS160 | MATα ade2-101 leu2-3,112 ura3-52 arg8::URA3 kar1-1 | ρ$^0$ | Steele et al. 1996 |
| YCy5051 | WT | MATa lys2 leu2-3,112 ura3-52 his3::HindIII arg8::hisG | ρ$^+$ | This study |
| YCy4179 | Δcox2::ARG8$^m$ | MATa lys2 leu2-3,112 ura3-52 his3::HindIII arg8::hisG | cox2::ARG8 m ρ$^+$ | This study |
| YCy5122 | epPCR clone E6 | MATa lys2 leu2-3,112 ura3-52 his3::HindIII arg8::hisG | cox2::ARG8 m ρ$^+$ | This study |
| YCy5123 | epPCR clone E8 | MATa lys2 leu2-3,112 ura3-52 his3::HindIII arg8::hisG | cox2::ARG8 m ρ$^+$ | This study |
| YCy5124 | epPCR clone C7 | MATa lys2 leu2-3,112 ura3-52 his3::HindIII arg8::hisG | cox2::ARG8 m ρ$^+$ | This study |
| YCy5125 | epPCR clone F7 | MATa lys2 leu2-3,112 ura3-52 his3::HindIII arg8::hisG | cox2::ARG8 m ρ$^+$ | This study |
| YCy5126 | epPCR clone F9 | MATa lys2 leu2-3,112 ura3-52 his3::HindIII arg8::hisG | cox2::ARG8 m ρ$^+$ | This study |

with hydrophobicity, both strategies only partially restored mitochondrial function. As such, identifying the limitations and improving the expression system itself is a key step to the successful improvement of allotopic expression. In such context, the tractability of baker's yeast *Saccharomyces cerevisiae* for both nuclear and mtDNA modification provides an excellent model to study the mechanism of human mitochondrial diseases and to explore new therapeutic strategies (Lasserre et al. 2015; Kucharczyk et al. 2019; Su et al. 2019; Ding et al. 2020).

In this work, we utilized engineering biology tools to improve the allotopic expression of the previously described *COX2-W56R* system (Supekova et al. 2010). A study conducted by Supekova et al. (2010) demonstrated that the single amino-acid substitution W56R in the first transmembrane helix (TMH1) of the allotopic Cox2p enabled a partial restoration of the respiratory growth of a yeast Δcox2::ARG8$^m$ deletion mutant. However, expression remained low, suggesting further adaptation may restore full mitochondrial function. Over the past decade, several studies have attempted to identify the *in vivo* fate of the allotopic Cox2-W56Rp, and how this expression system could be improved (Cruz-Torres et al. 2012; Rubalcava-Gracia et al. 2018; Rubalcava-Gracia et al. 2019; Nieto-Panqueva et al. 2024). Notably, one study suggested that decreasing *COX2-W56R* gene dosage by transferring the transcriptional unit from a high-copy vector to either a low-copy vector or to a genomic locus significantly improved respiratory growth of the Δcox2::ARG8$^m$ deletion mutant (Rubalcava-Gracia et al. 2019). However, previous studies were unable to isolate further protein sequence modifications that provide enhanced respiratory function, a challenge that we address here.

We used YeastFab assembly (Guo et al. 2015) to build a combinatorial expression vector library, varying the delivery vector, promoter, and MTS to thoroughly screen for optimal allotopic expression conditions. Further, we used error-prone PCR to generate novel adaptive mutations. Combining identified improvements, we demonstrated that an optimized *COX2* allotopic expression system confers a substantial improvement of respiratory growth of a Δcox2::ARG8$^m$ deletion mutant. While our findings require testing in human cell models to assess their translational value, this study demonstrates that yeast is a valuable biological model for exploring gene therapy approaches for mitochondrial diseases.

## Materials and methods
### Yeast strains, transformation, and culture
Biolistic transformation (Bonnefoy and Fox 2001) was used to deliver constructs containing either the wild-type (WT) *COX2* locus or the cox2::ARG8$^m$ deletion cassette to the mitochondria of the ρ$^0$ yeast strain DFS160 (strain YCy3894; Steele et al. 1996). Crossing between respective transformants of DFS160 and a strain carrying a mutant allele of the mitochondrial *COX2* gene (cox2-22) (strain NB64/YCy4097) was performed (Bonnefoy et al.

2001). This generated a strain with a restored WT *COX2* locus (strain YCy5051, referred to as WT) and a Δcox2::ARG8$^m$ deletion strain (strain YCy4179). Expression vectors were transformed into the obtained deletion strain using a LiOAc-based transformation protocol (Ito et al. 1983). Transformed cells were grown without shaking in synthetic complete media with dropouts corresponding to the used auxotrophy markers supplemented with 2% glucose. To perform spot tests, we spotted a 10-fold serial dilution of each strain, starting from a concentration of 1 OD$_{600nm}$ unit/mL on rich media (1% yeast extract and 2% peptone) supplemented with either 2% glucose or 2% glycerol and grew the cells at 30 degrees for the indicated time. A list of the yeast strains used in this study is given in Table 1.

### Plasmid construction
The sequence of the mitochondrial *COX2* gene from the S288C strain was recoded to match the *S. cerevisiae* nuclear genetic code using the SnapGene® software (from Dotmatics; available at snapgene.com; Nakamura et al. 2000). This sequence, along with the MTS from *Neurospora crassa* ATP synthase subunit 9 (sequences provided in Supplementary Table 1), was submitted for chemical synthesis (Twist Bioscience, South San Francisco, CA, USA). The MTS from the *QCR2* and *SCO2* genes were chemically synthesised as gBlocks by IDT (Leuven, Belgium). Other MTSs and promoter sequences were amplified from the yeast genome (see Supplementary Table 1 for the complete list of DNA sequences and primers used in this study). The Golden Gate-based YeastFab pipeline (Guo et al. 2015) was used with modified connectors for the rapid assembly of variant expression vectors, featuring a promoter, a MTS, an ORF and a terminator together in a yeast/bacteria shuttle vector. Assembled constructs were propagated in *Escherichia coli* TOP10 and purified using the E.Z.N.A.® Plasmid DNA Mini Kit I (Omega Bio-tek, Norcross, GA, USA) following the manufacturer's recommendations.

### Error-prone PCR, selection procedure and analysis of mutations additivity
We performed error-prone PCR (epPCR) using the GeneMorph II Random Mutagenesis Kit (Agilent) following the manufacturer's protocol to obtain an average of 4 mutations per kb throughout the single-copy *OXA1* MTS and the whole *COX2-W56R* gene. The mutant sequence library was cloned into a low-copy vector containing an *ICL1* promoter and *ADH1* terminator and transformed into *E. coli* TOP10 competent cells. Around 2000 clones were pooled for plasmid purification and transformation of the library into the Δcox2::ARG8$^m$ deletion strain (YCy4179). Transformed cells were plated on synthetic complete media with dropouts corresponding to the used auxotrophy markers and supplemented with 0.2% glucose and 2% glycerol to drive increased growth in cells able to efficiently metabolise glycerol. 93 clones from around 2000 yeast

colonies were identified as exhibiting a growth advantage and were transferred to a 96-well culture plate alongside the WT strain (YCy5051), the Δ*cox2::ARG8^m* deletion strain (YCy4179), and the Δ*cox2::ARG8^m* deletion strain (YCy4179) expressing the non-engineered *COX2-W56R* allotopic construct. Cultures were grown until saturation for 3 days at 30°C without shaking and plated on YP + Glucose 2% or Glycerol 2% (Supplementary Fig. 1a) using a ROTOR replicator robot (Singer Instruments, UK). After an additional 3 days of growth at 30°C, plates were imaged and colony sizes assessed using a PhenoBooth (Singer Instruments, UK; Supplementary Fig. 1a). The 9 largest colonies were selected (Supplementary Fig. 1b) and a spot assay was performed (Fig. 3b). Vectors were extracted from the 5 best growing clones and subjected to whole plasmid sequencing (Plasmidsaurus, Eugene, OR, USA). Identified mutations were mapped on the Cox2-W56Rp 3D model generated using SWISS-MODEL, available on the ExPASy Server (Fig. 3e; Waterhouse *et al.* 2018). The additive effect of mutations identified in TMH2 was explored further by combining identified mutations using the epPCR clone E6 plasmid (the centromeric POT2-URA3 vector containing the *ICL1* promoter, one copy of the *OXA1* MTS, the engineered *COX2-W56R-I190V* gene from epPCR clone E6 and *ADH1* terminator) and Golden Gate assembly to combinatorially introduce our panel of mutations. Constructs were then transformed into the *cox2* deletion strain for spot assay analysis of the conferred growth benefit (Fig. 3b).

## Analysis of mitochondrial crude extract

Strains for mitochondria extraction were inoculated in static liquid pre-cultures in 25 mL of synthetic liquid media with dropouts corresponding to the used auxotrophy markers and supplemented with 2% glucose. After 24 h, cells were harvested and inoculated in 1 L of YP + glycerol 2%, grown overnight at 30°C, harvested in exponential phase of growth, and subjected to enzymatic mitochondrial isolation, as previously described (Timón-Gómez *et al.* 2020). Oxygen consumption rates (OCR, using a Clark electrode from Hansatech, UK) and ATP synthesis rates were measured using freshly isolated and osmotically protected mitochondria (0.15 mg.mL$^{-1}$) in respiration buffer (10 mM Tris-maleate pH 6.8, 0.3 mM EGTA, 0.65 M sorbitol, and 3 mM potassium phosphate) at 30°C, as previously described (Rigoulet and Guerin 1979). Final concentrations of reaction substrates were as follows: 4 mM NADH, 150 μM ADP (for OCR assay), 750 μM ADP (for ATP synthesis assay), 12.5 mM ascorbate (Asc), 1.4 mM N,N,N,N,-tetramethyl-p- phenylenediamine (TMPD), 4 μM carbonyl cyanide-m-chlorophenyl hydrazone (CCCP), and 3 μg.mL$^{-1}$ oligomycin. ATP production was determined using a luciferin/luciferase assay (ATP determination kit, ThermoFischer). The CN-PAGE analyses were conducted as previously described.(Paumard *et al.* 2002; Ding *et al.* 2020) Briefly, 200 μg of isolated mitochondria were resuspended in 20 μL of extraction buffer (2% digitonin (Sigma), 1 mM EGTA, 150 mM potassium acetate, 12% glycerol, 30 mM HEPES, 2 mM 6-aminocaproic acid, protease inhibitor cocktail (Roche); pH 7.4), incubated for 30 min on ice and centrifuged for 30 min at 4°C. Supernatants were combined with 2.25 μL of loading buffer (0.01% Ponceau S dye, 750 mM 6-amino-caproic acid) and ran on a NativePAGE™ 3–12% Bis-Tris gel. Following migration, gels were incubated in either Coomassie staining solution, NADH dehydrogenase activity buffer (5 mM Tris, 1 mg.mL$^{-1}$ nitroblue tetrazolium, 0.5 mM NADH, pH 7.4), Complex IV activity buffer (5 mM Tris, 0.5 mg.mL$^{-1}$ diaminobenzidine, 1 mg.mL$^{-1}$ Cytochrome *c*, pH 7.4) or F1/FO ATPase activity buffer (0.27 M glycine, 35 mM Tris, 14 mM magnesium sulfate,

8 mM ATP, 0.2% lead nitrate, 0.1% Trito X-100, pH 8.4). Finally, after Coomassie staining, gels were blotted onto PVDF membranes and subjected to immunodetection using primary antibodies against Cox2p (Abcam, ref ab110271, RRID: AB_10858117, dilution 1:2500), Cox1p (Abcam, ref ab110270, RRID:AB_10863346, dilution 1:333), and Porin (ThermoFischer, ref 459500, RRID: AB_2532239, dilution 1:1000). Detection was done using an anti-mouse secondary antibody coupled to peroxidase (Sigma, ref A9044, 1:10000, RRID:AB_258431) and the Clarity Western ECL substrate kit (Biorad, ref 1705060S). The Western blot raw data and calculated ratios are given in Supplementary Table 4.

## Miscellaneous procedures

Following transformation of the *cox2* deletion strain (strain YCy4179) with the different expression constructs, transformants were grown overnight at 30°C in static synthetic liquid media with dropouts corresponding to the used auxotrophy markers and supplemented with 2% glucose. Cultures were then transferred to 20 mL of YPGly 2% at a starting OD$_{600nm}$ of 0.1.mL$^{-1}$ and grown overnight at 30°C with shaking. Finally, cells were harvested in exponential growth phase for whole-cell OCR analysis, or snap frozen for total protein extraction. Following extraction, total proteins were subjected to SDS-PAGE (Laemmli 1970) and western blotting. The abundance of the Cox2p (Abcam, ref ab110271, dilution 1:2500, RRID:AB_10858117) was normalized in each condition to that of Pgk1p (Abcam, ref ab113687, dilution 1:5000, RRID: AB_10861977). Then, for each condition, the Cox2p abundance is expressed as a percentage of the WT level. The western blot raw data and calculated ratios are given in Supplementary Table 4. Sequence comparison between *S. cerevisiae* (UniProt ID: P00410) and *Homo sapiens* (UniProt ID: P00403) was conducted using the Clustal Omega tool available on the EMBL-EBI website (Goujon *et al.* 2010; Sievers *et al.* 2011) For microscopy analysis, cells were imaged with an inverted epifluorescence microscope (Nikon Eclipse TE2000U) equipped with an 100× immersion objective and a standard FITC filter.

## Statistical analyses

Each experiment was performed using 3 individual biological replicates per condition. Prior to comparing results between groups, normality and homogeneity of variance was checked using the Shapiro–Wilk's test and the Levene test ($\alpha = 0.05$). If both null hypotheses were accepted, a parametric one-way ANOVA corrected by a Dunnett's test was performed, otherwise, we used a Kruskal–Wallis test corrected by a Mann–Whitney test ($\alpha = 0.05$). The statistical analyses were performed using the software RStudio version 4.2.3; given *P*-values are the corrected ones.

## Results

### Choice of expression system to improve

To select a suitable starting point from which to optimise the *COX2-W56R* allotopic expression system, we initially worked to reproduce the strategy previously described by Supekova and collaborators (Supekova *et al.* 2010). To this end, we used YeastFab assembly (Guo *et al.* 2015) to build high-copy expression vectors analogous to those previously described, as well as low-copy vector-encoded counterparts (Fig. 1a). The different expression vectors were transformed into our Δ*cox2::ARG8^m* deletion mutant (strain YCy4179) and subjected to a spot test to evaluate their ability to restore respiratory growth (Fig. 1b). Interestingly, while our results obtained using a high-copy vector mirror those described in both studies (Supekova *et al.* 2010; Rubalcava-Gracia *et al.*

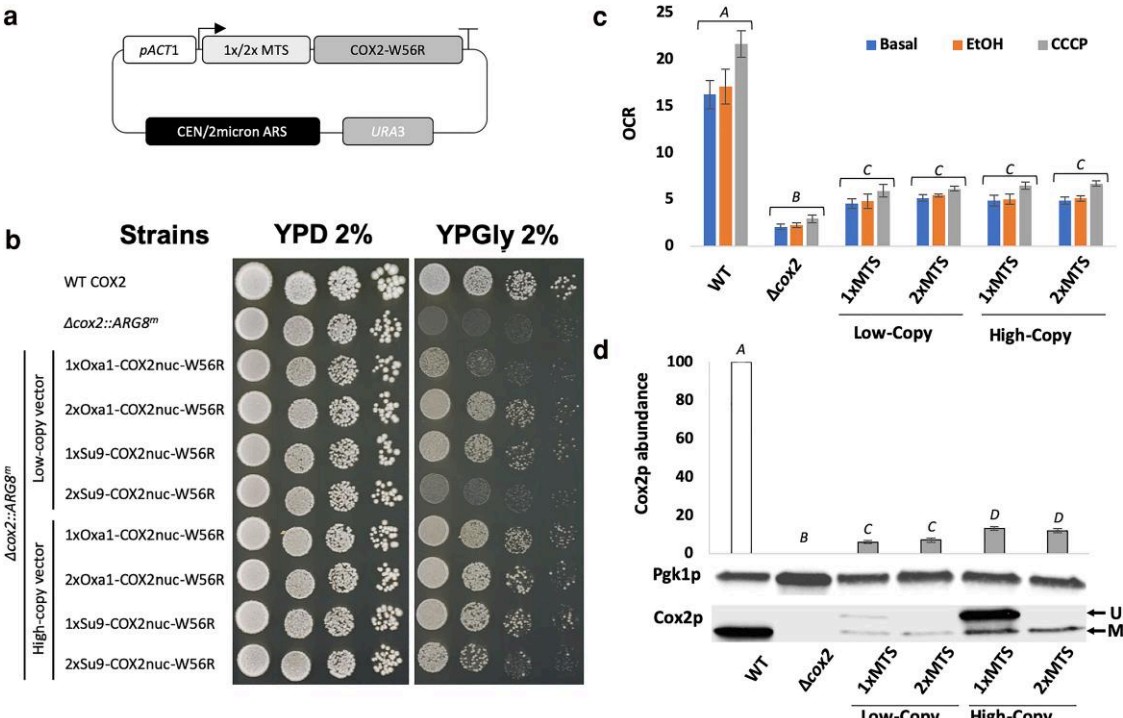

**Fig. 1.** a) Schematic representation of the design used to replicate expression vectors from Supekova *et al.* 2010 using the YeastFab assembly. b) Spot test presenting the growth benefit conferred by the different *COX2-W56R* allotopic constructs in the Δ*cox2::ARG8^m* strain, when hosted on a low- or a high-copy plasmid after 3 days at 30°C. c) Oxygen consumption rate (OCR, nmol.min$^{-1}$.OD$_{600nm}$ unit$^{-1}$, mean ± SD, $n = 3$) measured in whole cells at the basal level and following addition of Ethanol (EtOH) and CCCP. d) Western blot-based quantification of the Cox2p abundance relative to the Pgk1p control [mean (expressed as a percentage of the WT level) ± SD, $n = 3$]. *A, B, C* and *D*: Denotes a significant difference between the conditions ($P < 0.05$). U, unprocessed; M, mature.

2019), switching to a low-copy vector did not bring any benefit, but rather reduced fitness in respiratory conditions. This may be due to our use of the *ACT1* promoter instead of the *PGK1* promoter, as previously described (Supekova *et al.* 2010; Rubalcava-Gracia *et al.* 2019). In addition, we observed a greater respiratory growth benefit when combining the *COX2-W56R* gene with two copies of the *OXA1* MTS instead of just one, in agreement with a previous study (Supekova *et al.* 2010).

Following this, we sought to investigate the influence of plasmid copy number on whole cell oxygen consumption rate (OCR) and relative Cox2p abundance. OCR (Fig. 1c) was measured by resuspending cells in rich media devoid of a carbon source (basal respiration), and upon addition of ethanol alone (EtOH-stimulated respiration) or in combination with the uncoupling agent CCCP (CCCP-stimulated respiration). As expected, in our Δ*cox2::ARG8^m* deletion strain the OCR was severely diminished with only a low background OCR. The observed background OCR level was likely related to alternative metabolic pathways [i.e. sterol biosynthesis (Parks and Adams 1978) or unsaturated fatty acids synthesis (Rose and Harrison 1989)] rather than mitochondrial respiration. We observed a partial restoration of OCR when cells expressed the allotopic *COX2-W56R* gene. Interestingly, the degree of OCR recovery appeared to be largely similar for all constructs, irrespective of the vector and MTS copy number.

Western blot analysis confirmed the absence of Cox2p in our Δ*cox2::ARG8^m* deletion strain, and we saw a twice higher amount of mature Cox2p in high-copy conditions compared with low-copy conditions (12% vs 6%, Fig. 1d). However, while the constructs with 2 *OXA1* MTS copies displayed stronger restoration of respiratory growth (Fig. 1b), the effect on mature Cox2p abundance was minimal. Instead, we observed decreased accumulation of

unprocessed allotopic Cox2p (Fig. 1c). These results mirror previous work (Rubalcava-Gracia *et al.* 2019), confirming the cytoplasmic accumulation of unprocessed allotopic Cox2p when associated with a single copy *OXA1* MTS. Dual copy *OXA1* MTS, therefore, likely aids the clearance of accumulated unprocessed form avoiding potentially stressful cytoplasmic accumulation.

## Modulating allotopic protein expression level improves respiratory growth

We observed that hosting the *COX2-W56R* gene on a high-copy vector granted an improved respiratory growth phenotype (Fig. 1b) in our Δ*cox2::ARG8^m* deletion strain (strain YCy4179) compared with a low-copy vector. Thus, we hypothesized that increasing the expression level of the *COX2-W56R* gene from a low-copy vector, by changing the constitutive *ACT1* promoter for a stronger promoter, could similarly improve mutant respiratory growth. Such an effect has been previously described (Rubalcava-Gracia *et al.* 2019). To identify the best promoter candidates, we built a library of expression vectors with a GFP reporter under the control of various promoters, and transformed these constructs in our BY4741 rho$^+$ strain. Cells were grown in rich media supplemented with either 2% glucose or 2% glycerol, with GFP fluorescence quantified at different timepoints (Fig. 2a). Promoter candidates conferring medium to low expression in fermentative conditions, but higher expression than *pACT1* in respiratory conditions were selected for further investigation. Those regulated promoters were chosen to avoid unnecessary Cox2p expression during fermentative growth. Three candidate promoters were selected: *pADH1*, *pJEN1* and *pICL1,* conferring a 1.5-fold, 2-fold, or 3.5-fold higher expression level in respiratory conditions compared with *pACT1,* respectively (Fig. 2a).

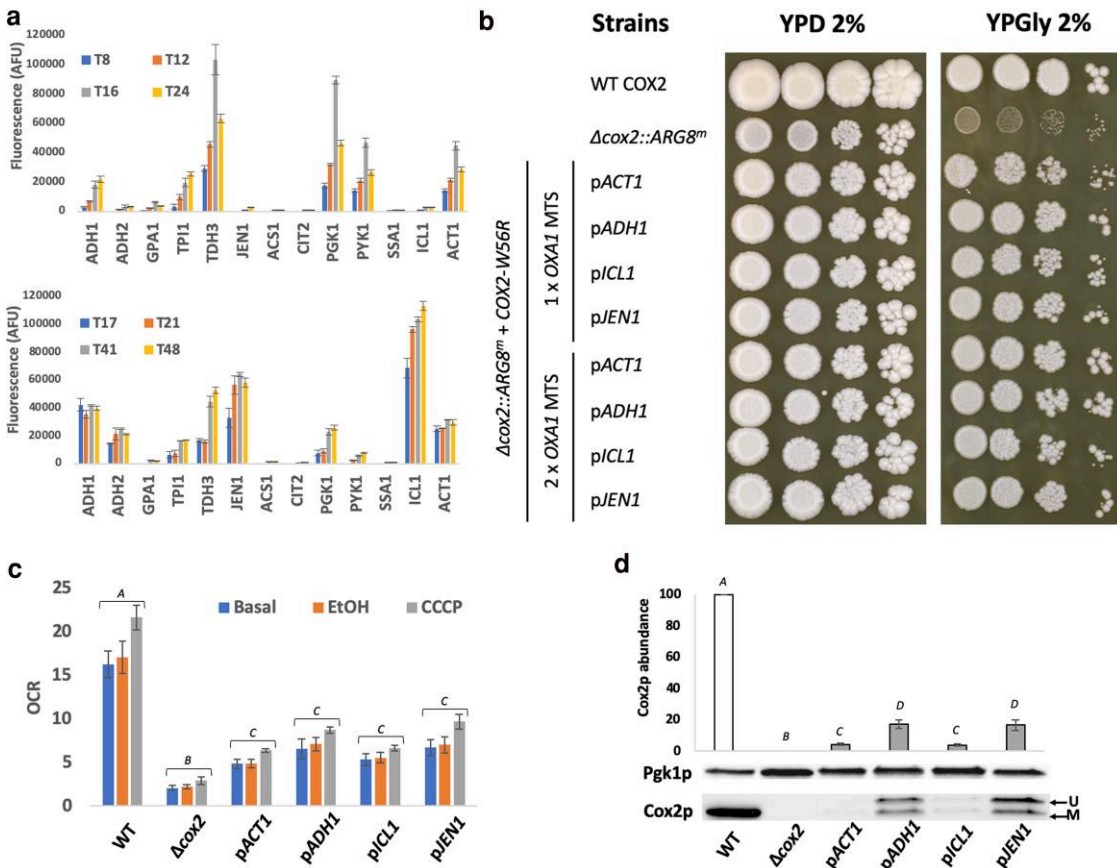

**Fig. 2.** a) Normalized GFP signal (GFP signal/OD$_{600nm}$) measured in the BY4741 strain expressing the different GFP constructs in glucose-containing rich media (2% glucose for 24 hours, upper chart) or ethanol-containing rich media (2% EtOH for 48 hours, lower chart) (n = 3). b) Spot test performed to assess the growth benefit conferred by the *COX2-W56R* allotopic construct expressed under the selected promoters after 7 days at 30°C (from *ACT1*, *ADH1*, *ICL1*, or *JEN1*). c) Oxygen consumption rate (OCR, nmol.min$^{-1}$.OD$_{600nm}$ unit $^{-1}$) measured in whole cells at the basal level and following addition of Ethanol (EtOH) and CCCP. d) Western blot-based quantification of the Cox2p abundance relative to the control Pgk1p [mean (expressed as a percentage of the WT level) ± SD, n = 3]. AFU, arbitrary fluorescence unit. *A*, *B*, *C* and *D*: denotes a significant difference between the conditions (P < 0.05). U, unprocessed; M, mature.

Our updated promoter panel was used to build new *COX2-W56R* constructs and a further spot assay was performed to evaluate the recovery of respiratory growth in deletion mutants (Fig. 2b). We observed a slight increase in expression level and improvement of respiratory growth recovery in cells carrying pADH1-driven constructs. This effect was amplified when 2 copies of the *OXA1* MTS were used. Interestingly, increasing the expression level using pJEN1 did not confer a greater recovery of respiratory growth. Where 2 copies of the *OXA1* MTS were used, pJEN1 constructs appeared to show reduced fitness gain compared with pADH1 constructs (Fig. 2b). Similarly, the use of the strongest promoter, pICL1, did not provide a fitness gain over pACT1.

As the best functional range between the different promoters was observed with a single copy of the *OXA1* MTS, we decided to focus on these variants to monitor the OCR on whole cells (Fig. 2c) and the relative *Cox2-W56Rp* abundance (Fig. 2d). When we used pADH1 and pJEN1, the OCR increased by 40% and the abundance of the mature Cox2-W56Rp was markedly increased by around 4-fold in comparison to the condition using pACT1. Surprisingly, the use of pICL1 did not increase Cox2-W56Rp abundance or OCR. Interestingly, while the amount of mature Cox2-W56Rp was similar with pADH1 and pJEN1, the amount of accumulated unprocessed protein was higher with the latter. It has previously been reported that the Cox2-W56Rp accumulates at the mitochondrial periphery, (Rubalcava-Gracia *et al.* 2019)

and may explain the loss of fitness if this accumulation is associated with overloading of the protein import machinery. Thus, at this stage, to obtain a better recovery of the mitochondrial function, we hypothesized that the protein sequence itself must be engineered.

## Engineering the allotopic protein sequence to improve respiratory growth

Supekova and collaborators showed a partial functional complementation of the Δ*cox2::ARG8$^m$* deletion through allotopically expressed *COX2* gene requires changes to its amino acid sequence. Using random mutagenesis by error-prone PCR, the authors identified a crucial amino acid substitution in the first TMH (TMH1), W56R, which causes a reduction of the TMH1 hydrophobicity. However, our results demonstrated that Cox2-W56R protein is still poorly adapted to allotopic expression as: (1) only low levels of mature Cox2-W56R allotopic protein are detected and (2) a large fraction of the unprocessed form is accumulated and might impose a stress on cells. Even though the addition of a second copy of *OXA1* MTS could reduce the accumulation of the unprocessed form, no benefit was observed on the accumulation level of mature Cox2-W56Rp (Fig. 1d).

As such, we decided to further engineer the Cox2-W56Rp sequence using error-prone PCR, including one copy of the *OXA1* MTS along with the *COX2-W56R* gene, and sub-cloned the generated

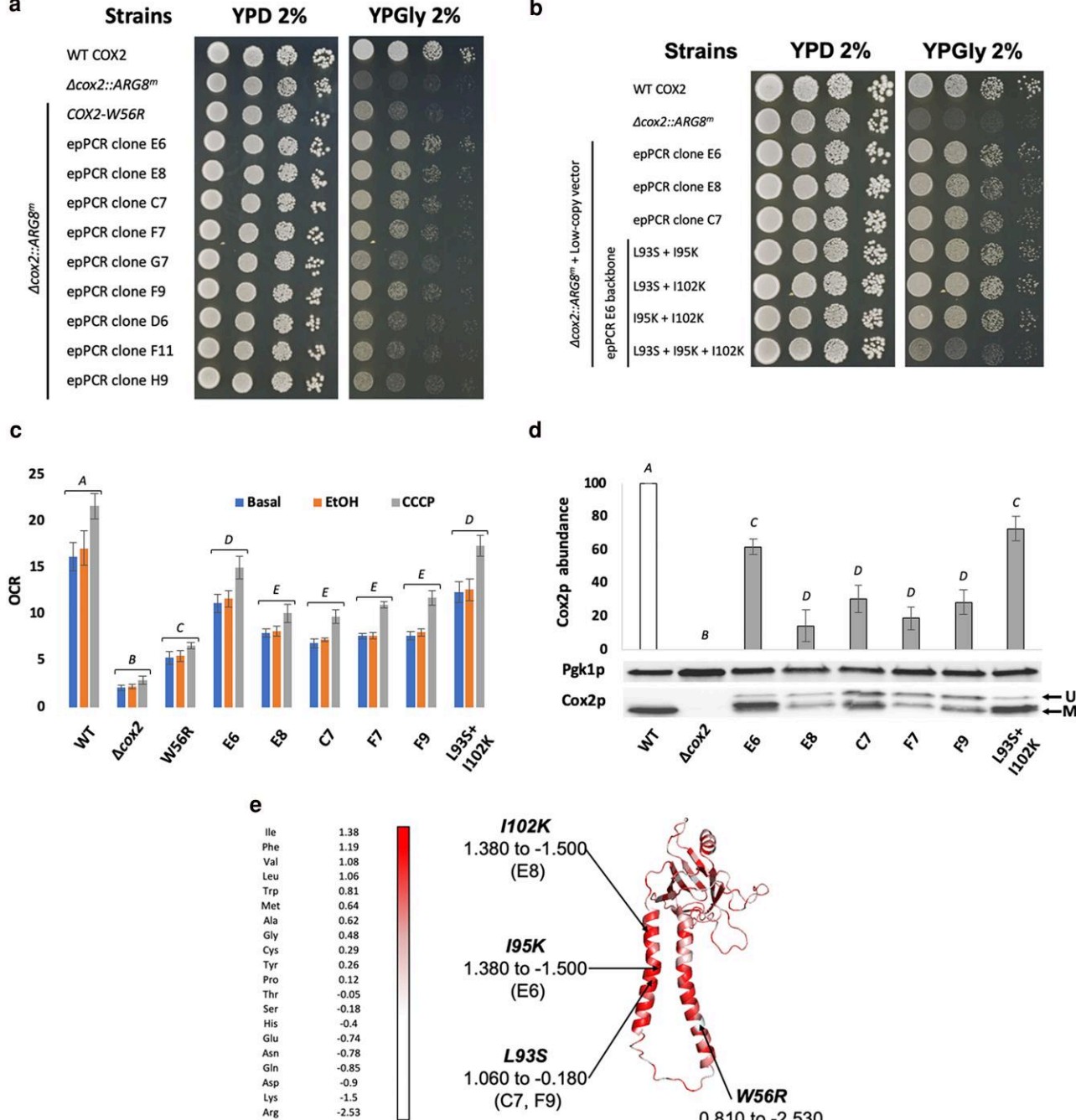

**Fig. 3.** a) Spot test of the 9 isolated best growing epPCR clones, grown for 3 days at 30°C. b) Spot assay testing the additivity of the mutations identified in the mutational hot spot in TMH2, grown for 4 days at 30°C. c) Oxygen consumption rate (OCR, $nmol.min^{-1}.OD_{600nm}$ unit $^{-1}$) measured in whole cells at the basal level and following addition of ethanol (EtOH) and CCCP. d) Western blot-based quantification of Cox2p abundance relative to the control Pgk1p [mean (expressed as a percentage of the WT level) ± SD, $n = 3$]. e) Annotation of the functional mutations identified in the protein sequence in the isolated epPCR clones E6, E8, C7, and F9 and their corresponding switch in hydrophobicity on the Cox2p protein structure generated using the SWISS-MODEL tool on the ExPASy Server and coloured based on amino acids hydrophobicity (red meaning hydrophobic and white hydrophilic) using PyMol. A, B, C, D, and E: denotes a significant difference between the conditions ($P < 0.05$). U, unprocessed; M, mature.

library in a low-copy vector (the centromeric POT2 vector; Guo *et al.* 2015), placing the gene under the control of p*ICL1*. We decided to use this promoter, as it conferred strong expression in respiratory condition but was also associated with suboptimal respiratory growth recovery. We reasoned this would allow easy detection of favourable mutations in selective conditions. Using the procedure described in the methods section, we isolated 9 epPCR clones with the highest assessed growth recovery and performed a spot test for confirmation (Fig. 3a). The obtained results demonstrated that the ranking based

on the colony size in selective media allowed to efficiently identify epPCR clones carrying favourable mutations. Indeed, 5 of the isolated top 9 epPCR clones (E6, E8, C7, F7, and F9) were growing better than the parental strain (strain YCy4179 expressing the non-engineered *COX2-W56R* gene) in respiratory conditions. Their plasmids were extracted, propagated in bacteria, and sequenced to identify mutations (listed in Supplementary Table 1; Fig. 3e). Clones C7 (strain YCy5124) and F9 (strain YCy5126) carried the same mutations. We observed that 2 mutations were present in all

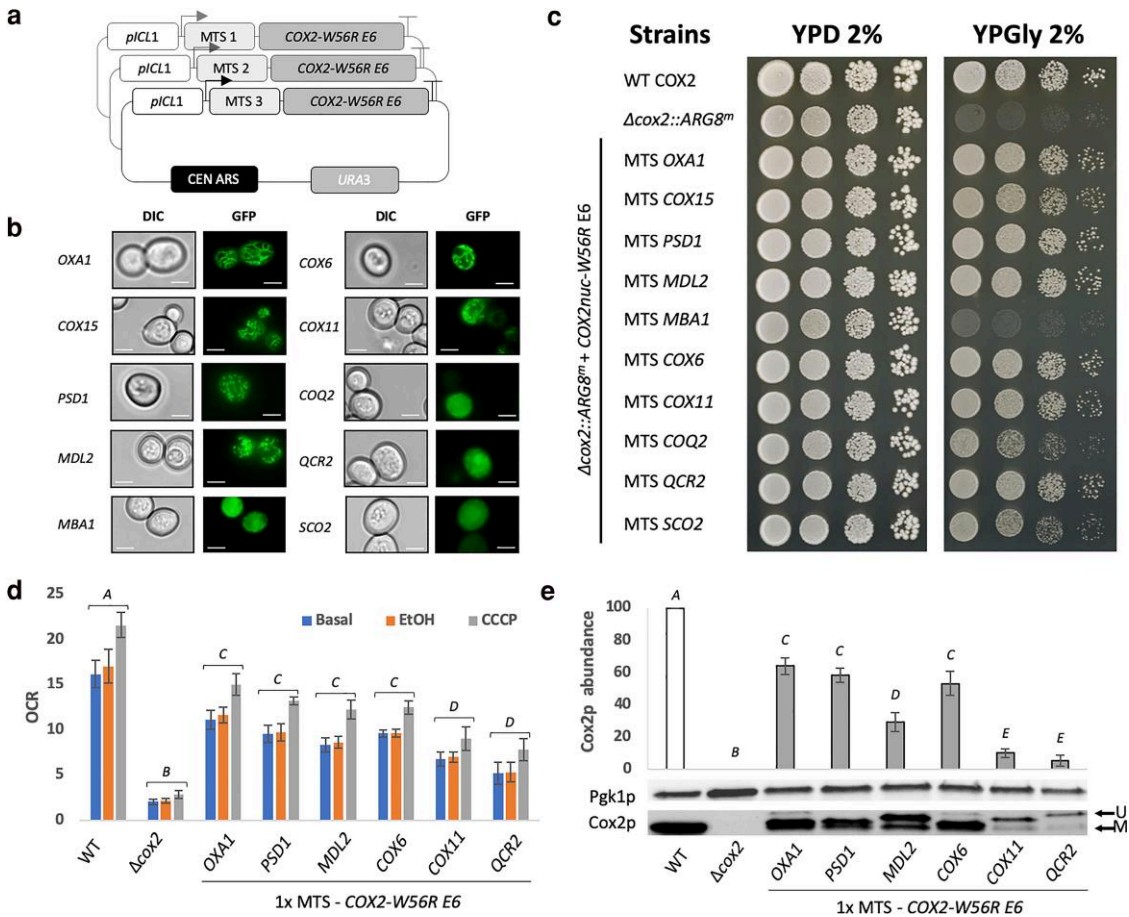

**Fig. 4.** a) Schematic representation of constructs for the MTS screening using the sub-optimal allotopic construct from epPCR clone E6. b) Fluorescence imaging of the BY4741 rho[+] strain expressing the different MTS-GFP constructs. Scale bar = 5 μm. c) Spot test presenting the respiratory growth benefit conferred by the screened MTSs as compared to OXA1 MTS after 4 days at 30°C. d) Oxygen consumption rate (OCR, nmol.min$^{-1}$.OD$_{600nm}$ unit $^{-1}$) measured in whole cells at the basal level and following addition of ethanol (EtOH) and CCCP. e) Western blot-based quantification of the Cox2p abundance relatively to the control Pgk1p [mean (expressed as a percentage of the WT level) ± SD, $n = 3$]. A, B, C, D, and E: denotes a significant difference between the conditions ($P < 0.05$). U, unprocessed; M, mature.

the clones: an A to C in position 21 and an A to G in position 709. While the former is silent and located in the MTS, the second is a missense mutation and produces an isoleucine to valine substitution (I190V) in the intermembrane space domain (IMS). To identify the origin of the 2 mutations found in all selected clones, we sequenced 4 clones from a naïve epPCR library. All 4 clones contained both mutations. We observed these mutations were present in a small fraction in the template and were unexpectedly enriched during the error-prone PCR, potentially explaining their presence in all the isolated clones. In addition to these mutations, multiple beneficial mutations have been identified in the TMH2, with 3 substitutions (I95K, I102K and L93S, Supplementary Table 2) responsible for drastic changes in hydrophobicity (Fig. 3e and Supplementary Fig. 2). Finally, the epPCR clone F7 did not have any mutations in the TMH2, but instead a threonine to arginine substitution in the MTS (T10R, Supplementary Table 2) and a leucine to methionine substitution in the IMS (L210M, Supplementary Table 2), both causing a slight decrease in hydrophobicity (Fig. 3e and Supplementary Fig. 2). To investigate the role of the identified mutations, each was individually reintroduced into the COX2-W56R gene and assessed for its ability to enhance respiratory growth in the Δcox2::ARG8$^m$ deletion mutant (strain YCy4179; Supplementary Fig. 1c). We found that the I190V and L210M mutations had negligible effects compared to those impacting TMH2. Notably, the I95K mutation

provided the greatest improvement, while L93S and I102K mutations showed a modest benefit.

As the TMH2 mutations substantially improved respiratory growth, we hypothesized that an additive effect might be observed when combining the 3 amino acid substitutions. Thus, we created different combinations and performed a new spot test to compare the growth benefit as compared to the parental strains (Fig. 3b). While combining the 3 mutations together abolished the growth benefit they individually brought, the combination L93S + I102K appeared to slightly improve the respiratory growth relative to the individual mutations. We then measured the associated OCR in the isolated epPCR clones and the combination L93S + I102K (Fig. 3c) and observed a net improvement in all the strains as compared to the parental W56R strain, which reached about 70% of WT levels for epPCR clone E6 and the combination L93S + I102K. This improvement was also confirmed at the protein level, with a substantial increase in mature Cox2p accumulation along with a decrease in the abundance of the unprocessed form in the most adapted strains (Fig. 3d).

## Exploring the efficiency of different MTSs to improve allotopic expression

Intriguingly, during the process of random mutagenesis and evolution, we isolated a better growing epPCR clone (F7, strain

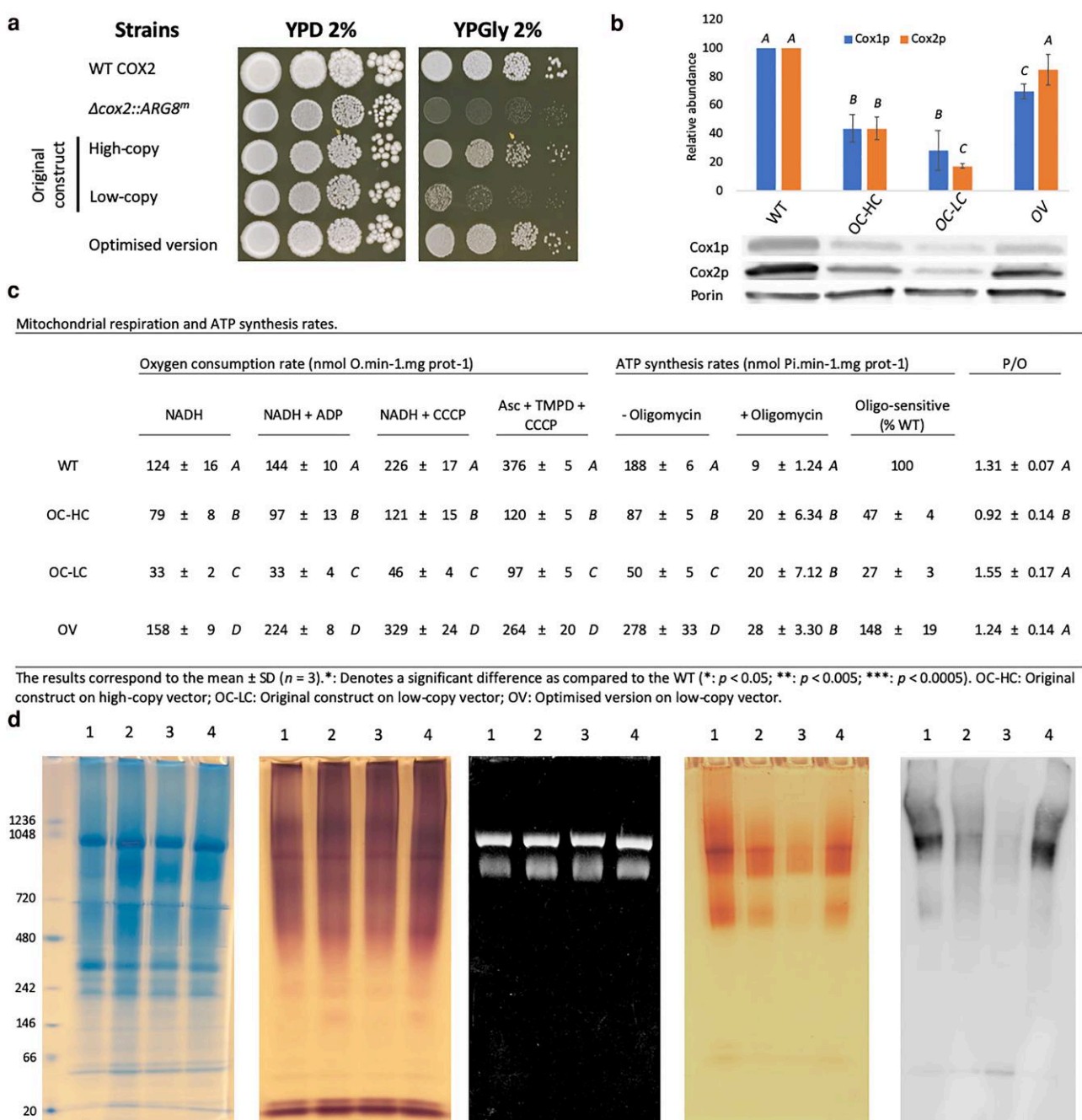

**Fig. 5.** a) Spot test comparing the growth benefit conferred by the initial construct from supekova and collaborators and the most optimized expression system generated in this study. Shown is growth after 4 days at 30°C. b) Western blot–based quantification of the Cox1p and Cox2p abundances relatively to the control Porin [mean (expressed as a percentage of the WT level) ± SD, $n = 3$]. c) Oxygen consumption rate (OCR, nmol.min$^{-1}$.OD$_{600nm}$ unit $^{-1}$) and ATP synthesis rate (nmol Pi.min$^{-1}$.mg prot$^{-1}$) measured in isolated mitochondria. d) CN-PAGE analysis performed on digitonin-solubilized mitochondria. Lane 1: WT; Lane 2: original construct on high-copy vector; Lane 3: original construct on low-copy vector; Lane 4: optimized version on low-copy vector. NADH DH: NADH dehydrogenase. A, B, C and D: denotes a significant difference between the conditions ($P < 0.05$). *Denotes a significant difference as compared to the WT (*$P < 0.005$; ***$P < 0.005$). OC-HC: previously described construct[5] hosted on a high-copy vector. OC-LC, previously described construct[5] hosted on a low-copy vector; OV, the optimized version from this study hosted on a low-copy vector.

YCy5125) that was carrying a T10R substitution in the MTS, causing a decrease in hydrophobicity (Fig. 3e and Supplementary Fig. 2) as well as the gain of positive charge (Supplementary Table 3). We thus wondered whether different MTSs with variable total positive charge could outperform the *OXA1* MTS. Nuclear relocation of the *COX2* gene has happened naturally, for instance in legumes. In these cases, *COX2* has acquired a long MTS (Adams *et al.* 1999;

Rodríguez-Salinas *et al.* 2012); so, we also tested whether a longer MTS could improve the allotopic expression of Cox2p. To this end, we chose previously described MTSs (Vögtle *et al.* 2009) with both a higher net positive charge and length as compared with that of *OXA1* These MTSs are known to target their protein to either the matrix or the inner mitochondrial membrane (Supplementary Table 3). First, we tested each of these MTSs for their ability to

properly direct GFP to mitochondria (Fig. 4b). We observed that most of the selected MTSs properly targeted the fluorescent protein to mitochondria except for those from the *MBA1, COQ2, QCR2,* and *SCO2* genes. We thus built new expression vectors using the *ICL1* promoter, the different MTSs, the *COX2* gene sequence from epPCR clone E6 and a low-copy vector (Fig. 4a) and transformed these constructs in our Δ*cox2::ARG8^m* deletion mutant. Then, we evaluated the growth benefit they conferred, but none of them displayed a better respiratory growth than that using *OXA1* MTS (Fig. 4b). At best, they conferred a similar growth phenotype (MTS from *PSD1, MDL2, COX6, COX11,* and *QCR2*). Similarly, the OCR measured in these strains (Fig. 4c), as well as the relative abundance of the mature Cox2p (Fig. 4d) never exceeded those observed with the *OXA1* MTS.

## Building optimized allotopic COX2 expression system

Since the main aim of this study was to shed new light on the critical adaptations required for successful *COX2* nuclear relocation, we attempted to assemble all the gathered insights to generate the best allotopic *COX2* expression system. We first explored the benefit of combining the TMH2 L93S + I102K mutations with that observed in the MTS (T10R referred to as MTS *OXA1* F7) and the IMS (L210M) of the epPCR clone F7, but none of these combinations significantly improved respiratory growth (Supplementary Fig. 3a). Thus, we conserved the expression system using the native *OXA1* MTS and the TMH2 L93S + I102K mutations and tested the effect of swapping the promoter and the number of *OXA1* MTS copies (Supplementary Fig. 3b). Surprisingly, we observed that the addition of a second copy of the *OXA1* MTS did not improve respiratory growth, unlike initially observed for the poorly adapted *COX2-W56R* version (Fig. 1b and 2c). Regarding the change of promoter, spot tests indicated a slight improvement of respiratory growth when the transcriptional unit contained a single *OXA1* MTS copy and was placed under the control of the *ADH1* promoter, but this was not confirmed when the growth of the strains was monitored in liquid culture (Supplementary Fig. 3c). We thus decided to use the *ICL1* promoter with 1 or 2 copies of the *OXA1* MTS along with the *COX2-W56R-L93S-I102K* gene to perform a final test comparing the benefit of using either a low- or high-copy vector (Supplementary Fig. 3d). We confirmed that adding a second copy of the *OXA1* MTS brought no additional benefit nor did hosting the transcriptional unit on a high-copy vector. As such, we could identify the optimal construct within the explored parameter space: combining the engineered *COX2-W56R-L93S-I102K* gene with a single *OXA1* MTS under the control of the *ICL1* promoter and encoded on a low-copy vector. We then compared the respiratory growth benefit conferred by the optimized version (referred to as OV) to constructs expressing the *COX2-W56R* gene with one copy of *OXA1* MTS under the control of the *ACT1* promoter, replicating the functionally best expression system described by Supekova *et al.* (2010) (Fig. 5a). Respective reference constructs were made both with a high-copy (referred to as OC-HC) and a low-copy vector (referred to as OC-LC). We observed that our optimized version improved the fitness relative to these constructs, both under fermentative and respiratory conditions.

To complete this study, we decided to perform a biochemical characterization on mitochondria isolated from these strains. To this end, we grew the strains overnight in respiratory conditions (rich media + 2% glycerol) and isolated mitochondria from cells in exponential phase of growth. We then measured the OCR in the presence of different substrates: NADH alone (state 3); combined with ADP (state 4); following the addition of the proton

ionophore carbonyl cyanide m-chlorophenylhydrazone (CCCP, uncoupled maximal respiration); or using ascorbate + TMPD to assess the Complex IV specific activity. In addition, ATP synthesis rates were measured using NADH and ADP with or without the addition of the ATP synthase inhibitor oligomycin (Fig. 5c). When cells expressed the original construct on a low-copy vector, the OCR was about 20–25% of that in the WT, while using a high-copy vector increased the OCR to about 60% of the WT level. However, complex IV activity increased more modestly, from 20 to 30% of the WT level. In contrast, our optimized version restored OCR to a WT level, and the Complex IV activity reached up to 70% of the WT level. Similarly, the optimized version restored rates of ATP synthesis to a substantially higher degree than the original constructs.

Interestingly, isolated mitochondria from cells expressing our optimized version displayed a Complex IV activity restored up to 70% of the WT level (Fig. 5c) and thus we wondered whether the intramitochondrial level of the allotopic Cox2p protein could be the limiting step to full restoration of the Complex IV activity. We assayed relative abundances of Cox2p and the endogenous mitochondrial Cox1p (Fig. 5b) and observed that the accumulation of both proteins in each condition is similar and corroborates the measures of Complex IV activity (Fig. 5c). As intramitochondrial Cox2p abundance of the optimized version was nearly restored to a WT level, we investigated whether the mutated allotopic protein affects the efficiency of assembly of respiratory complexes, in particular, the formation of supercomplexes (SCs). To this end, isolated mitochondria were digitonin-solubilized (2 g/g protein) and subjected to CN-PAGE followed by *in gel* activity assessment (Fig. 5d). The NADH dehydrogenase and Complex IV activities were assessed to probe the efficiency of SCs assembly, while the F1/F0 ATPase activity and Coomassie staining were used as controls. First, the *in gel* NADH dehydrogenase activity pattern demonstrated that in all the experimental conditions, the electron transfer chain (ETC) complexes were able to form SCs. *In gel* activity and immunoblotting anti-Cox2p patterns confirmed the ability of Complex IV to form SCs in all the experimental conditions, though with an overall decreased abundance, corroborating observations obtained from isolated mitochondria (Fig. 5c). We could thus confirm that our optimized version supports the formation of SCs.

## Discussion

In this work, we harnessed multiple engineering biology approaches to refine the process of nuclear relocation of mitochondrial genes in yeast, focusing on the previously described *COX2-W56R* allotopic expression system (Supekova *et al.* 2010). To this end, we adapted YeastFab assembly, initially designed to build standard transcriptional units for metabolic engineering in yeast, to enable the integration and screening of diverse MTS. This framework facilitated the optimization of expression conditions, tuning promoter strength, sequence and repeat number of the MTS, and copy number of the expression system. We observed that fine-tuning of these expression parameters afforded only a moderate effect on *COX2-W56R* allotopic expression. Instead, engineering the protein sequence improved the level of mature Cox2p and cytochrome c oxidase accumulation (up to 70% of the WT level, Fig. 3d, 4b, and 4d) and decreased the cytoplasmic accumulation of non-matured form (Fig. 3d).

Interestingly, we observed that several favourable mutations were concentrated in the second TMH, resulting in a significant decrease in hydrophobicity (Fig. 3c and Supplementary Fig. 3). These

adjustments in TMH composition result in the modification of the average apparent free energy of membrane insertion ($\mu\Delta G_{app}$), probably favouring the TIM23-mediated mitochondrial import of the allotopic protein (Botelho *et al.* 2011, 2013; Nieto-Panqueva *et al.* 2023, 2024). It has been proposed that the W56R mutation increases the $\mu\Delta G_{app}$ of the TMH1, resulting in its translocation in the matrix, while the lower $\mu\Delta G_{app}$ of TMH2 triggers its retention by the TIM23 translocon and the lateral sorting of the allotopic Cox2-W56Rp (Nieto-Panqueva *et al.* 2023). We thus propose that the newly discovered adaptive mutations (I95K, I102K and L93S) increase the $\mu\Delta G_{app}$ of TMH2, which may favour the "conservative-sorting" import pathway instead of the "stop-transfer" pathway (Mahlke *et al.* 1990; Bohnert *et al.* 2010). Consequently, the matrix-delivered allotopic protein can be inserted in the inner mitochondrial membrane by the Oxa1p insertase following the same route as the naturally mitochondria-encoded Cox2p. These results confirm that adjusting the TMH composition of allotopic transmembrane mitochondrial proteins is a key factor in a successful adaptation to nuclear relocation (Popot and de Vitry 1990; Claros *et al.* 1995; Daley *et al.* 2002; Supekova *et al.* 2010; Bietenhader *et al.* 2012; Nieto-Panqueva *et al.* 2023). The last step of this optimization focused on testing different MTSs, but none of the candidate MTSs yielded a better phenotype than the *OXA1* MTS. To note, although the screening here did not yield an MTS with better performance, such a screening had substantially improved allotopic expression of a hydrophobic protein such as Atp6p in a human expression system (Chin *et al.* 2018).

We then explored the biochemical properties of mitochondria isolated from cells expressing our optimized *COX2* allotopic expression system. Surprisingly, the measured biochemical properties of these isolated mitochondria were restored to a WT level (OCR and ATP synthesis rates, Fig. 5c), despite a slight growth defect in respiratory conditions (Fig. 5a and Supplementary Fig. 3c). When we measured the specific activity of the Complex IV (Fig. 5c), we noticed that it was restored to 70% of the WT level and >2-fold and 3-fold higher than in conditions expressing the original construct on a high- and low-copy vector, respectively. Interestingly, when we measured the Cox2p intramitochondrial level, it was restored to a near-WT level (85% of the WT level, Fig. 5b) while the Cox1p abundance was about 70% of the WT level. Importantly, the biogenesis of Complex IV heavily relies on Cox1p, whose synthesis, assembly, and maturation are tightly regulated. Any disruption in this process triggers a selective degradation of the Cox1 subunit (Mick *et al.* 2011; Khalimonchuk *et al.* 2012; McStay *et al.* 2013) In the context of Cox2-W56Rp allotopic expression, it has been proposed that the low efficiency of assembly of the allotopic protein within the Complex IV resulted in decreased Cox1p abundance (Cruz-Torres *et al.* 2012; Rubalcava-Gracia *et al.* 2018). Thus, by substantially improving the intra-mitochondrial level of the allotopic Cox2p, we could also partially restore the Cox1p level and ultimately the abundance of mature Complex IV.

Interestingly, while the biochemical properties of isolated mitochondria from cells expressing our optimized version appeared restored, the respiratory growth of these cells was not fully recovered (Fig. 5a, Supplementary Fig. 3c). We confirmed that this was not due to a defect in Complex IV biogenesis nor in the capability of ETC complexes to form supercomplexes (Fig. 5c). Thus, we propose that despite a substantial improvement of the protein import and sorting, the Complex IV activity/accumulation is not fully restored, and cells present an initially reduced growth rate. However, as observed in liquid respiratory media, this growth defect was recovered after 33 h (Supplementary Fig. 3c). As such,

despite a delay in the proper expression, import and assembly of the allotopic protein, the accumulation of functional protein complexes enabled recovery of mitochondrial respiratory function over time.

Previous work to understand the fate of Cox2-W56Rp, and how to improve its expression *in vivo*, has proposed that a decreased *COX2-W56R* gene dosage is sufficient to substantially improve allotopic expression (Cruz-Torres *et al.* 2012; Rubalcava-Gracia *et al.* 2018; Rubalcava-Gracia *et al.* 2019). However, when adapting the *COX2* gene for allotopic expression in human cell models, engineering the protein sequence may provide better insights into the limiting factor to a successful nuclear relocation. To explore whether our results could potentially be applied to a human cell model, we aligned the human Cox2p sequence against its yeast counterpart (Supplementary Fig. 4). We observed that the initial substitution at position W56 in yeast is a tyrosine in human Cox2p (Y40), an amino acid with a mild hydrophobicity due to its hydroxyl group. In addition, 2 out of the 3 TMH2 residues found mutated in our study (L93 and I95) are well conserved in humans (L73 and L75). The third beneficial mutation in TMH2 (I102K in yeast) is already a highly hydrophilic arginine in human (R82). Strikingly, the IMS substitution common to all the epPCR clones isolated in this study, I190V, is also a valine in the corresponding position in human Cox2p. We explored the benefit of these mutations by allotopically expressing a yeast codon-optimized version of the human *MT-CO2* gene in our $\Delta cox2::ARG8^m$ deletion mutant (Supplementary Fig. 4b and 4c). However, none of the tested conditions lead to a significant restoration of the respiratory metabolism of the yeast mutant strain, probably due to the interplay of different chaperones and/or assembly factors between both organisms (Watson and McStay 2020). We propose that a future experimental relocation of *MT-CO2* in human cells should focus on engineering the amino acid sequences in the 2 transmembrane stretches to increase their $\mu\Delta G_{app}$, and particularly, testing substitutions at positions L73 and L75.

## Data availability

Strains and plasmids are available upon request. The authors affirm that all data necessary for confirming the conclusions of the article are present within the article, figures, and tables.

Supplemental material available at GENETICS online.

## Acknowledgments

The authors would like to thank Sara Isenc Le Digarcher for contributing in generating the OCR data on whole cells presented in this paper during her training. We thank Gabrielle David and Dr. Joshua James for proof-reading this manuscript. Finally, the authors would like to thank Dr. Jean-Paul di Rago, Dr. Déborah Tribouillard-Tanvier, and Prof. Lars Steinmetz for sharing their expertise on the topic of allotopic expression in yeast.

## Funding

This work was supported by a Wellcome Sanger Institute Associate Faculty Award, UK Biotechnology and Biological Sciences Research Council (BBSRC) grants BB/M005690/1, BB/P02114X/1, and BB/W014483/1; a Volkswagen Foundation "Life? Initiative" Grant (Ref. 94 771); an Engineering and Physical Sciences Research Council (EPSRC) Fellowship EP/V05967X/1; and a European Research Council (ERC) Consolidator Award EP/Y024753/1 to Y.C.

## Conflicts of interest

The author(s) declare no conflict of interest.

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
