## [Peer Review File · Genetics]

A new set of mutations in the second transmembrane helix of the Cox2p-W56R massively improves its allotopic expression in *Saccharomyces cerevisiae*.

Kewin Gombeau, Stefan Hoffmann, and Yizhi Cai

NOTE: The reviews and decision letters are unedited and appear as submitted by the reviewers.

In extremely rare instances and as determined by a Senior Editor or the EIC, portions of a review may be redacted. If a review is signed, the reviewer has agreed to no longer remain anonymous.

The review history appears in chronological order.

Review Timeline:

Submission Date:	2024-07-02
Editorial Decision:	2024-07-30
Resubmission Received:	2024-12-13
Editorial Decision:	2025-01-19
Revision Received:	2025-02-20
Accepted:	2025-02-23

July 30, 2024

GENETICS-2024-307248

A new set of mutations in the second transmembrane helix of the Cox2p-W56R massively improves its allotopic expression in *Saccharomyces cerevisiae*.

Dear Dr. Cai:

Three experts in the field have reviewed your manuscript, and I have read it as well. While your manuscript is not currently acceptable for publication in GENETICS, we would welcome a substantially revised manuscript. All reviewers have comments and concerns to be addressed in a revised manuscript. You can read their reviews at the end of this email.

We look forward to receiving your revised manuscript. Please let the editorial office know approximately how long you expect to need for revisions.

Upon resubmission, please include:

1. A clean version of your manuscript;
2. A marked version of your manuscript in which you highlight significant revisions carried out in response to the major points raised by the editor/reviewers (track changes is acceptable if preferred);
3. A detailed response to the editor's/reviewers' feedback and to the concerns listed above. Please reference line numbers in this response to aid the editor and reviewers.

Your paper will likely be sent back out for review.

Additionally, please ensure that your resubmission is formatted for GENETICS
<https://academic.oup.com/genetics/pages/general-instructions>

Follow this link to submit the revised manuscript: Link Not Available

Sincerely,

Junbiao Dai
Associate Editor
GENETICS

Approved by:
Meera Sundaram
Senior Editor
GENETICS

Reviewer #1 (Comments for the Authors (Required)):

Background: The COX2 gene is encoded in the mitochondrial genome, therefore rescue of a cox2 mutant would normally require introduction of the WT copy into the mitochondrial genome, a technically difficult process. The alternative is to introduce the COX2 gene into the nuclear genome, with modifications such that the protein product is efficiently localized to mitochondria. This "artificial relocation" to the mitochondria of a mitochondrial gene product artificially expressed in the nucleus is termed allotopic expression. This approach has the potential to rescue defective respiratory function caused by mutations acquired in mitochondrial genes (for example as seen in a variety of human mitochondrial disorders), and is therefore of broad interest.

The goal of this study was to shed light on the critical adaptations required for successful allotopic expression of the COX2 mitochondrial protein. The starting point was a plasmid encoded COX2-W56R mutant gene containing a mitochondrial targeting signal (MTS) that is expressed in the nucleus, that had previously been shown to rescue a cox2 null mutation (Supekova, PNAS 2010). Supekova showed that the W56R mutation, identified in a mutagenesis screen, acted in cis to facilitate protein transport into the mitochondria thereby facilitating complementation of a cox2 null mutant, albeit not to full WT levels.

In this paper, the authors conducted a follow-on mutagenesis screen, and expression vector optimization effort, starting with the COX2-W56R gene, with the goal of identifying additional cis-acting missense mutations that improve mitochondrial protein expression levels and complementation (4 independent isolates were identified, each carrying multiple missense mutations).

They also tested the effects of various promoters, MTS sequences, and vector copy number, to optimize expression and promote improved complementation of mitochondrial respiration and cell growth.

In conclusion to the optimization efforts, the abstract states, "Importantly, the most beneficial amino acid substitutions found in the second transmembrane helix (L93S and I102K) are conserved residues in the corresponding positions of human COX2 (L73 and L75) and we propose that mirroring these changes could help improve allotopic Cox2p expression in human cells." It was not clear to me that these specific residues were unambiguously shown to elicit the desired cis-effects, so this will have to be clarified if this statement is to stand.

This paper represents an enormous amount of effort. The final optimized plasmid expression vector (ICL1 promoter - Oxa1 MTS - COX2ORF with 4 missense mutations-low copy vector) resulted in an improved oxygen consumption rate from 60% (using a vector analogous to the Supekova expression construct) to a WT level, as well as improved complex IV activity from 30% to 70%, and increased rates of ATP synthesis.

Overall, the paper describes a detailed roadmap for testing a set of variables to optimize allotopic expression of a mitochondrial gene in yeast. The authors successfully applied this approach to optimizing allotopic expression of COX2 in yeast. The optimized COX2 expression plasmid can be used in future functional studies. Whether the specific mutations found in the yeast COX2 gene would elicit similar effects by introduction into the corresponding human gene is unknown, but could be tested, and if successful might be incorporated into strategies for rescuing human disease mutations that map to COX2. The extent to which these results could directly guide the introduction of specific mutations in other mitochondrial genes to successfully promote allotopic expression is unknown, but provide good starting points.

Major point: All 4 independent clones that were selected from the PCR mutagenesis screen had 2 identical mutations (1 missense, 1 silent). This seems statistically impossible in a screen of 2000 clones at the mutation rate reported (especially the silent mutation that should not confer better growth and is therefore a "passenger" mutation). The library DNA needs to be carefully QC'd. See *** below.

Specific points:

Line 25: I think "directed evolution" (iterative rounds of selection, mutagenesis, and amplification to mimic natural selection) is not the correct term for this work. The term is used in several places in the manuscript.

Line 27, abstract, and elsewhere: "we massively increased the mitochondrial incorporation..." I am not sure massively is the correct term. Perhaps significantly is better.

There are large differences between the various constructs made in this work, but the benchmark for improved expression would be the W56R expression plasmid described in the 2020 Supekova paper, which has the Act1 promoter, single OXA1 or SU9 MTS, COX2ORF, in a high copy vector (i.e., moderate promoter, high copy, see fig 6 of 2020 PNAS paper).

Line 30-34, abstract: As mentioned above, I could not find data demonstrating that "the most beneficial amino acid substitutions were found in the second transmembrane helix (L93S and I102K)". I may be missing something, but I think all 4 independent clones that were sequenced had multiple missense mutations. All 4 independent clones had the identical I190V missense mutation (suggesting this residue is the functional variant), as well as additional missense mutations (presumably passenger mutations?). I think the L93S and I102K mutations were engineered (added) into the E6 isolate that carries 2 missense mutations (I190V and I95K) to create a quadruple mutant. It is not clear to me that there is a significant difference between the E6 double mutant and the quadruple mutant in the growth, OCR, and Cox2p abundance assays in Fig 3 B, C, D. Were any of these mutations tested and analyzed as single mutations?

Line 109: Change "massive" to "significant". The benchmark would be the Supekova 2020 construct.

Line 156-159: It is not clear why the ICL1 promoter was chosen for the mutagenesis library. I see that it is stated in results (lines 368-372), so perhaps this is not an issue for experimental methods section.

Lines 159-171: 2000 transformants were screened and 93 candidates (identified by visual inspection) transferred to 96-well format, and rescored on glucose rich or glycerol containing media. In FigS1, the Phenobooth heat map of the 93 isolates and controls grown on glucose rich media (lower left panel) demonstrated considerable variation, several candidates growing better than the WT control.

First- Does this lower left panel data make sense?

Second- The "nine largest colonies" (heat map on glycerol, lower right panel, were further spot tested (Fig S1B, Fig 3A (not3B)). Later it was shown that C7 and F9 are sibs from the library, as they have the identical 3 mutations, 2 being missense, but in the Phenobooth re-analysis, C7 grows better than F9, suggesting apparent growth variation in independent transformants of the same plasmid due to other factor(s). In my opinion, it seems possible that initial transformants might be expected to exhibit differences in CEN-plasmid copy number at some frequency (some transformants may be derived from cells that took up two or three copies). This could explain the apparent difference between clones C7 and F9, for example. C7 and F9 also show minor

variation in Figure 3. In my opinion, recovery and retransformation of initial candidate plasmids, and testing several independent transformants, prior to sequencing, would have been advisable. Nonetheless, the authors were able to confirm that constructs identified in the initial screens performed better than the parental construct in subsequent experiments.

*** Third- The fact that all 4 independent clones each have the same two nucleotide substitutions (one missense and one silent) embedded in other mutation suggests that the original template for PCR may have had one or both of these mutations. Given this result, QC of the library DNA (extracted from E coli and that was used in the yeast transformation screen) should be reported. This could be addressed by sequencing 4 random isolates that did not appear to be growing better on the initial yeast transformation plate, or sequencing 4 random clones after transforming the library DNA back into E. Coli and analyzing random plasmid transformants.

If generated during PCR, the missense mutation in common to all clones could be inferred to be the functional variant, and the others inferred as passenger mutations (in the absence of functionally testing the variants in isolation). The variation across clones could be explained if the "passenger" mutations have secondary effects on the mutation in common. Alternatively, if the common mutation was in the PCR template, then one of the other missense mutations could be inferred as being the functional variant.

Lines 175-177: "Mutational hotspots were explored further". What criteria is being used to call a mutation a hotspot? In this context, they would have to be functional mutations, not passengers, to be meaningful. Given multiple missense mutations in each of the clones, some of them are likely passenger mutations.

Line 249: Heading phrase: "Identification of the best expression system to improve". Need to state what is improvement goal is being pursued here.

Line 275-277: "Interestingly, we observed a partial restoration of OCR when cells expressed the allotopic COX2-W56R gene, recovery which increased with the vector but not the MTS copy number." It is not clear to me whether the differences in OCR in Fig 1C are statistically significant when comparing across constructs with different MTS number or plasmid copy number. I may not understand.

(Similarly- Lines 301-303, same issue (significance in cross-comparison). "We observed that hosting the COX2-W56R gene on a high-copy vector, as compared to a low-copy vector, granted the strongest restoration of respiratory growth in our *cox2::ARG8m* 303 deletion strain (strain YCy4179)."

Line 286-288: "confirming the cytoplasmic accumulation of unprocessed allotopic Cox2p when associated with a single copy OXA1 MTS. Dual copy OXA1 MTS therefore likely aids mitochondrial import of our Cox2-W56Rp, avoiding potentially stressful cytoplasmic accumulation." The amount of mature Cox2 protein looks to be the same comparing one vs two MTS segments, but the amount of unprocessed COX2 is high with one MTS1 and undetectable with two MTS1. This doesn't seem to be increased import. Could it be that the unprocessed species is degraded more efficiently when it contains two MTS1, for example?

Lines 328-331: "As the best functional range between the different promoters was observed with a single copy of the OXA1 MTS, we decided to focus on these conditions to monitor the OCR on whole cells (Figure 2.C) and the relative Cox2-W56Rp abundance (Figure 2.D)." I don't understand this rationale, especially the meaning of best functional range. Wouldn't you want to start with the best performing construct and try to improve it?

Single MTS accumulates unprocessed to greater extent than double MTS. Also, double MTS improves low copy vector growth in glycerol and accumulates lower amounts of unprocessed COX2, which could be toxic.

Lines 368-372: "As such, we decided to further engineer the Cox2-W56Rp sequence using error prone PCR, including one copy of the OXA1 MTS along with the COX2-W56R gene, and sub-cloned the generated library in a low-copy vector (the previously described centromeric POT2 vector)(Guo et al. 2015) placing the gene under the control of pICL1." OK, rationale is to start with compromised phenotype in order to more easily detect missense mutations that improve performance. Do I understand this correctly?

Lines 385-389. Need to address the fact that two identical mutations were found in every clone (as described above).

Lines 396-398: "As the TMH2 hotspot mutations massively improved respiratory growth, we hypothesised that an additive effect might be observed when combining the three amino acid substitutions." Was there a statistically significant difference between E6 and E6 plus the two TMH2 mutations (growth)? Massively is the wrong descriptor.

Lines 406: Similarly: "This improvement was also confirmed at the protein level, with a massive increase in mature Cox2p accumulation along with a decrease in the abundance of the unprocessed form in the most adapted strains (Figure 3.D). Was there a statistically significant difference between E6 and E6 plus the two TMH2 mutations (protein)? Massive is the wrong descriptor.

Lines 482-487: "We then compared the respiratory growth benefit conferred by the optimised version (referred to as OV) from

this study as compared to the original construct described by Supekova and collaborators (Supekova et al. 2010) hosted on either a high-copy vector (latter referred to as OC-HC) or a low-copy vector (latter referred to as OC-LC) (Figure 5.A). We observed that our optimised version improved the fitness relative to the original constructs, both under fermentative and respiratory conditions."

Please clarify: The benchmark for improved expression would be the W56R expression plasmid described in the 2020 Supekova paper, which has the Act1 promoter, single OXA1 or SU9 MTS, COX2ORF, in a high copy vector (i.e., moderate promoter, high copy, see fig 6 of 2020 PNAS paper). Was the OC-HC plasmid analyzed here (Fig 5) identical to the Supekova construct, or a construct made in this work that has similar elements? If the latter, then the comparison needs rewording.

Line 597, line 614, and elsewhere: "massively" should be reworded.

Line 556-557: "Interestingly, we observed that several highly favourable mutations were concentrated in the second TMH" and Lines 571-573: "These results confirm that adjusting the TMH composition of allotopic transmembrane mitochondrial proteins is a key factor in a successful adaptation to nuclear relocation." and Final sentence 627-630: "we propose that an experimental relocation of MT-CO2 in human cells should focus on engineering the amino acid sequences in the two transmembrane stretches to increase their $\mu\Delta G_{app}$, and particularly, testing substitutions at positions L73 and L75."

As discussed above, were these mutations shown to encode highly favorable variants when tested in isolation?" If not, then these statements should be altered.

Reviewer #2 (Comments for the Authors (Required)):

In this brief investigative manuscript, Gombeau et al. describe how allotopic expression of Cox2 can be improved in budding yeast, starting from an already established mutation on Cox2, W56R. The authors have explored several ways, such as using one or two MTS, different promoters, or additional mutations on the Cox2 to optimize the expression. They have taken advantage of biochemical assays and oxygen consumption assays as a readout of the improved expression. Finally, the authors have tested their improved version for mitochondrial function. Given the potential of gene therapy approaches in treating mitochondrial diseases, there is an exciting future for understanding and developing tools to improve the allotopic expression of specific mitochondrial genes. The data presented in the paper is solid and convincing. However, the work represents an incremental conceptual advance and needs to provide more new insight that would make it of general interest to the readership of Genetics. Some additional comments are outlined below for the authors to consider

Minor queries:

1. Line 323: Do the authors really want to say pJEN1 or pICL1? The growth assay shown in Figure 2B would suggest the authors were trying to say ICL1. The growth assay on glycerol is intended to study fitness in respiratory conditions. However, it is hard to interpret, and it is hard to tell which one is growing better. So, I would recommend the authors repeat the assay and keep the plates for a longer time.
2. Growing yeasts on glycerol media will enhance lipid synthesis and thereby residual OCR from lipid metabolic pathways. So do the authors consider using a different fermentable carbon source for their assays?

Reviewer #3 (Comments for the Authors (Required)):

The authors utilized the previously published COX2-W56R system to overcome the respiratory defect caused by delta cox2::ARG8m. They first verified the experimental phenotype that has been reported in 2019 and then screened for the higher expression promoters which could replace the high-copy vector in low-copy vectors. Using an error-prone PCR-based method, they identified two point mutations that significantly rescued the respiratory phenotype under the ICL1 promoter. Ultimately, they developed an optimized allotopic COX2 expression system.

Major Points:

1. The authors should include an empty plasmid as a control in all spotting assays to demonstrate that the vector itself does not affect respiratory processes.
2. While the results are interesting and robust, the connection of this research to human mitochondrial disease and gene therapy seems tenuous.
3. In the promoter screening experiment, it is unclear why the authors chose ICL1 for the OCR test and epPCR for sequence optimization, given that figure 2a suggests TDH3 could also be a good candidate.
4. The statistical analysis of the OCR experiments shows significant differences between each mutant and the wild type.

However, it is important to clarify if there are differences between the various promoters, high or low copy vectors, and different evolved coding sequences.

Minor points:

- 1 It is unclear why the high copy plasmid could already rescue the delta *cox2::ARG8m* effectively, yet the authors opted to optimize the low copy vector by changing promoters rather than directly optimizing the coding sequence in the high copy vector.
- 2 In figure 2a, it is interesting that most highly expressing promoters peak at 18 hours, whereas ethanol peaks at 41 hours. An explanation for this observation would be helpful.
- 3 What happens when the optimized final system is expressed in a high copy vector? Would it show even better performance?
- 4 Including mRNA expression levels of strains expressing COX2 rescue vectors would strengthen the findings.
- 5 The U/M in the western blot figure should be clearly specified in the figure legend.
- 6 There are some typos, please correct. For example: P21 line 442 at best, "thy"....

Associate Editor Comments:

We sincerely appreciate the thoughtful and constructive reviews of our manuscript provided by all three referees. Their valuable insights and suggestions were instrumental in improving the manuscript. We are also grateful to the reviewers for recognizing the significant effort put into this study, and for acknowledging the robustness and comprehensive nature of the presented data.

Based on their specific feedbacks, we have revised the manuscript as detailed here.

Reviewer 1

1.1 *In conclusion to the optimization efforts, the abstract states, "Importantly, the most beneficial amino acid substitutions found in the second transmembrane helix (L93S and I102K) are conserved residues in the corresponding positions of human COX2 (L73 and L75) and we propose that mirroring these changes could help improve allotopic Cox2p expression in human cells." It was not clear to me that these specific residues were unambiguously shown to elicit the desired cis-effects, so this will have to be clarified if this statement is to stand.*

We thank the reviewer for pointing this out. To evidence that these specific residues are causative for the phenotype, we opted to characterize the effects of beneficial mutations individually. To this end, we cloned COX2-W56R variants with the following additional individual mutations: I190V, L210M, L93S, I95K or I102K (see Figure S1). I190V and L210M did not seem to bring any benefit over the parental COX2-W56R. Out of sampled mutations in TMH2, I95K conferred the greatest benefit, consistent with the growth phenotype of respective isolated epPCR clones (see Figure 3). Notably, while the mutations L93S and I102K individually had a minor effect on the respiratory growth (See Figure S1C and D), the combination of both markedly improved respiratory growth of the COX2 deletion strain (see Figure 3 and 5).

To highlight these results, we have added the following text in the manuscript:

- Line 492-497: "To investigate the role of the identified mutations, each was individually reintroduced into the COX2-W56R gene and assessed for its ability to enhance respiratory growth in the $\Delta\text{cox2}::\text{ARG8m}$ deletion mutant (Figure S1.C). We found that the I190V and L210M mutations had negligible effects compared to those impacting TMH2. Notably, the I95K mutation provided the greatest improvement, while the L93S and I102K mutations showed a modest benefit "

1.2 *All 4 independent clones that were selected from the PCR mutagenesis screen had 2 identical mutations (1 missense, 1 silent). This seems statistically impossible in a screen of 2000 clones at the mutation rate reported (especially the silent mutation that should not confer better growth and is therefore a "passenger" mutation). The library DNA needs to be carefully QC'd.*

The fact that all 4 independent clones each have the same two nucleotide substitutions (one missense and one silent) embedded in other mutation suggests that the original template for PCR may have had one or both of these mutations. Given this result, QC of the library DNA (extracted from E coli and that was used in the yeast transformation screen) should be reported.

This could be addressed by sequencing 4 random isolates that did not appear to be growing better on the initial yeast transformation plate, or sequencing 4 random clones after transforming the library DNA back into E. Coli and analyzing random plasmid transformants.

If generated during PCR, the missense mutation in common to all clones could be inferred to be the functional variant, and the others inferred as passenger mutations (in the absence of functionally testing the variants in isolation). The variation across clones could be explained if the "passenger" mutations have secondary effects on the mutation in common. Alternatively, if the common mutation was in the PCR template, then one of the other missense mutations could be inferred as being the functional variant.

The reviewer raises an excellent point. We previously had attributed the observation that all sampled clones shared two mutations (a missense and a silent one) to a potential beneficial impact of the missense mutation, leading to its enrichment together with the silent mutation as a passenger mutation. However, sequencing four *E. coli* clones derived from an error-prone PCR all showed both mutations, despite being from a naïve library. Re-analysing the sequencing trace of the plasmid used as a template suggested it to contain a mixture of the expected sequence and one containing the mutations in questions. For reasons unknown, the species with the mutations appear to be strongly enriched during library generation. To determine whether this mutation is beneficial, we generated and tested the missense mutation I190V individually (see previous point) and found no phenotypical difference as compared to the COX2-W56R version.

We have added the following section to the main text explaining that the two mutations found in every sampled clone originate from the epPCR template.

- Line 482-487: “To identify the origin of the two mutations found in all selected clones, we sequenced four clones from a naïve epPCR library. All four clones contained both mutations. We observed these mutations were present as a small fraction in the template and were unexpectedly enriched during the error-prone PCR, explaining their presence in all the isolated clones.”

1.3 *Line 25: I think "directed evolution" (iterative rounds of selection, mutagenesis, and amplification to mimic natural selection) is not the correct term for this work. The term is used in several places in the manuscript.*

We have replaced all instances of the term “directed evolution” with “random mutagenesis and selection”.

1.4 *Line 27, abstract, and elsewhere: "we massively increased the mitochondrial incorporation..." I am not sure massively is the correct term. Perhaps significantly is better. There are large differences between the various constructs made in this work, but the benchmark for improved expression would be the W56R expression plasmid described in the 2020 Supekova paper, which has the Act1 promoter, single OXA1 or SU9 MTS, COX2ORF, in a high copy vector (i.e., moderate promoter, high copy, see fig 6 of 2020 PNAS paper).*

We have replaced the use of “massively” in all instances, including in the title, by “substantially”, which arguably is a more fitting term.

1.5 *Line 30-34, abstract: As mentioned above, I could not find data demonstrating that "the most beneficial amino acid substitutions were found in the second transmembrane helix (L93S and I102K)". I may be missing something, but I think all 4 independent clones that were sequenced had multiple missense mutations. All 4 independent clones had the identical I190V missense mutation (suggesting this residue is the functional variant), as well as additional missense mutations (presumably passenger mutations?). I think the L93S and I102K mutations were engineered (added) into the E6 isolate that carries 2 missense mutations (I190V and I95K) to create a quadruple mutant. It is not clear to me that there is a significant difference between the E6 double mutant and the quadruple mutant in the growth, OCR, and Cox2p abundance assays in Fig 3 B, C, D. Were any of these mutations tested and analyzed as single mutations?*

Initially, we had not tested the mutations in question individually. Admittedly, characterizing single mutations allows gauging their individual contributions and potential epistatic interactions. Thus, we created and tested single mutations in question in the parental W56R background, please see our response to reviewer comment 1.1. Consequently, by combining L93S and I102K (in presence of the silent mutation in the MTS and I190V) we can confidently conclude that the observed benefit is conferred by the reduced hydrophobicity in TMH2.

1.6 *Line 109: Change "massive" to "significant". The benchmark would be the Supekova 2020 construct.*

Following this reviewer’s suggestions, instances of describing functional improvements as “massive” have been changed (see our response to 1.4).

1.7 *Line 156-159: It is not clear why the ICL1 promoter was chosen for the mutagenesis library. I see that it is stated in results (lines 368-372), so perhaps this is not an issue for experimental methods section.*

This promoter confers a low expression in fermentation condition and a strong expression in respiratory condition. We have reworded the explanation in the results section for a clearer explanation for choosing this particular regulated promoter. Please also refer to our response to comment 1.14.

1.8 *Lines 159-171: 2000 transformants were screened and 93 candidates (identified by visual inspection) transferred to 96-well format, and rescored on glucose rich or glycerol containing media. In FigS1, the Phenobooth heat map of the 93 isolates and controls grown on glucose rich media (lower left panel) demonstrated considerable variation, several candidates growing better than the WT control.*

First- Does this lower left panel data make sense?.

The lower left panel is a heatmap presenting the size of the colonies formed on the YPD plate. The total spread of colony sizes on YPD is rather small, and the heatmap scaled to represent these relatively small

size differences. On first glance the colonies all appear to have about the same size, but thanks to the heatmap it can be inferred that there seem to be some positional effects (e.g. an edge effect).

Second- The "nine largest colonies" (heat map on glycerol, lower right panel, were further spot tested (Fig S1B, Fig 3A (not3B). Later it was shown that C7 and F9 are sibs from the library, as they have the identical 3 mutations, 2 being missense, but in the Phenobooth re-analysis, C7 grows better than F9, suggesting apparent growth variation in independent transformants of the same plasmid due to other factor(s). In my opinion, it seems possible that initial transformants might be expected to exhibit differences in CEN-plasmid copy number at some frequency (some transformants may be derived from cells that took up two or three copies). This could explain the apparent difference between clones C7 and F9, for example. C7 and F9 also show minor variation in Figure 3. In my opinion, recovery and retransformation of initial candidate plasmids, and testing several independent transformants, prior to sequencing, would have been advisable. Nonetheless, the authors were able to confirm that constructs identified in the initial screens performed better than the parental construct in subsequent experiments.

The reviewer raises a valid point, with recovery and re-transformation of candidate plasmids being a methodologically stringent way to assess conferred phenotypes. Here, the Phenobooth screening was intended as an easily scalable first-line screening. Due to the nature of this assay, there is considerable measurement noise (e.g. due to aforementioned positional effects). Top candidates were validated by more robust spot assays, with the rank of candidates being largely in good agreement with the Phenobooth screening. The difference of C7 and F9 in the screening, however, is most likely spurious.

1.9 *Lines 175-177: "Mutational hotspots were explored further". What criteria is being used to call a mutation a hotspot? In this context, they would have to be functional mutations, not passengers, to be meaningful. Given multiple missense mutations in each of the clones, some of them are likely passenger mutations*

As explained in point 1 and presented in Figure S1.C, the missense mutation I190V has negligible effect as compared to those identified in TMH2, which happened to be functional mutations. As such, we consider it relevant to use the expression "Mutational hotspots were explored further" when discussing the role of these mutations.

1.10 *Heading phrase: "Identification of the best expression system to improve". Need to state what is improvement goal is being pursued here.*

We thank the reviewer for highlighting this point. To avoid confusion for the readers, we have changed the heading phrase to "Identification of the expression system to improve" so the readers can better appreciate the selection procedure described in this paragraph.

1.11 *Line 275-277: "Interestingly, we observed a partial restoration of OCR when cells expressed the allotopic COX2-W56R gene, recovery which increased with the vector but not the MTS copy number." It is not clear to me whether the differences in OCR in Fig 1C are statistically significantly when comparing across constructs with different MTS number or plasmid copy number. I may not understand.*

None of the differences between high and low-copy and MTS number are significant. Accordingly, we rephrased the section as follows: "We observed a partial restoration of OCR when cells expressed the allotopic COX2-W56R gene, which interestingly was apparently independent of both the vector and the MTS copy number."

(Similarly- Lines 301-303, same issue (significance in cross-comparison). "We observed that hosting the COX2-W56R gene on a high-copy vector, as compared to a low-copy vector, granted the strongest restoration of respiratory growth in our Δ cox2::ARG8m 303 deletion strain (strain YCy4179)."

We thank the reviewer for highlighting these points. Indeed, there are no statistical differences in OCRs presented in Figure 1, and we modified the sentence in the text as follow: "Interestingly, we observed a partial restoration of OCR when cells expressed the allotopic COX2-W56R gene, recovery which was not impacted by the vector nor the MTS copy number."

Regarding the second point, we do understand the confusion and modified the sentence as follow: "We observed that hosting the COX2-W56R gene on a high-copy vector, as compared to a low-copy vector, granted an improved respiratory growth phenotype (Figure 1.B) in our Δ cox2::ARG8^m deletion strain (strain YCy4179)."

1.12 *Line 286-288: "confirming the cytoplasmic accumulation of unprocessed allotopic Cox2p when associated with a single copy OXA1 MTS. Dual copy OXA1 MTS therefore likely aids mitochondrial import of our Cox2-W56Rp, avoiding potentially stressful cytoplasmic accumulation." The amount of mature Cox2 protein looks to be the same comparing one vs two MTS segments, but the amount of unprocessed COX2 is high with one MTS1 and undetectable with two MTS1. This doesn't seem to be increased import. Could it be that the unprocessed species is degraded more efficiently when it contains two MTS1, for example?*

We do agree with the reviewer's comment and modified the sentence accordingly: "Dual copy OXA1 MTS therefore likely aids the clearance of accumulated unprocessed form avoiding potentially stressful cytoplasmic accumulation".

1.13 *Lines 328-331: "As the best functional range between the different promoters was observed with a single copy of the OXA1 MTS, we decided to focus on these conditions to monitor the OCR on whole cells (Figure 2.C) and the relative Cox2-W56Rp abundance (Figure 2.D)." I don't*

understand this rationale, especially the meaning of best functional range. Wouldn't you want to start with the best performing construct and try to improve it?

Single MTS accumulates unprocessed to greater extent than double MTS. Also, double MTS improves low copy vector growth in glycerol and accumulates lower amounts of unprocessed COX2, which could be toxic.

We opted to start with the expression system conferring a smaller benefit for respiratory growth (one rather than two OXA1 MTS), as we reasoned it would allow easier detection of better growing clones. We reasoned that expression of COX2-W56R with a single copy of the OXA1 MTS, where respiratory growth was poor relative to two copies of the OXA1 MTS, probably led to a majority of the alltopic protein aggregating in the cytoplasm and creating a stress. We hypothesized that the main driver for the cytoplasmic aggregation is the hydrophobicity of the alltopic protein, and thus, we would be able to identify adaptive mutations preventing cytoplasmic aggregation and/or improving import and sorting in this context.

1.14 *Lines 368-372: "As such, we decided to further engineer the Cox2-W56Rp sequence using error prone PCR, including one copy of the OXA1 MTS along with the COX2-W56R gene, and sub-cloned the generated library in a low-copy vector (the previously described centromeric POT2 vector)(Guo et al. 2015) placing the gene under the control of pICL1." OK, rationale is to start with compromised phenotype in order to more easily detect missense mutations that improve performance. Do I understand this correctly?*

Indeed, pICL1 was chosen based on the assumption that it would facilitate detection of beneficial mutations. We rephrased the explanation to make the rationale clearer:

"As such, we decided to further engineer the Cox2-W56Rp sequence using error-prone PCR, including one copy of the OXA1 MTS along with the COX2-W56R gene, and sub-cloned the generated library in a low-copy vector (the centromeric POT2 vector)(Guo et al. 2015) placing the gene under the control of pICL1. We decided to use this promoter, as it conferred strong expression in respiratory condition, but was associated with suboptimal respiratory growth recovery. We reasoned this would allow easy detection of favourable mutations in selective conditions."

1.15 *Lines385-389. Need to address the fact that two identical mutations were found in every clone (as described above).*

See comment 1.1.

1.16 *Lines 396-398: "As the TMH2 hotspot mutations massively improved respiratory growth, we hypothesised that an additive effect might be observed when combining the three amino acid substitutions." Was there a statistically significant difference between E6 and E6 plus the two TMH2 mutations (growth)? Massively is the wrong descriptor.*

In the ANOVA of OCR and quantitative western blot, there was no significant difference between E6 and E6 with L93S+I102K. However, growth in the spot assay of the latter was visibly better than that of E6 alone, and both its OCR and Cox2p abundance were numerically higher than that of E6.

We have replaced “massively” by “substantially”.

1.17 *Lines 406: Similarly: "This improvement was also confirmed at the protein level, with a massive increase in mature Cox2p accumulation along with a decrease in the abundance of the unprocessed form in the most adapted strains (Figure 3.D). Was there a statistically significant difference between E6 and E6 plus the two TMH2 mutations (protein)? Massive is the wrong descriptor.*

Please note that this line refers not to the comparison between E6 and E6 plus the two TMH2 mutations, but the comparison to the parental W56R COX2 variant. That difference is significant for both OCR and Cox2p abundance. We rephrased the preceding section to make that clearer.

1.18 *Lines 482-487: "We then compared the respiratory growth benefit conferred by the optimised version (referred to as OV) from this study as compared to the original construct described by Supekova and collaborators(Supekova et al. 2010) hosted on either a high-copy vector (latter referred to as OC-HC) or a low-copy vector (latter referred to as OC-LC) (Figure 5.A). We observed that our optimised version improved the fitness relative to the original constructs, both under fermentative and respiratory conditions."*

Please clarify: The benchmark for improved expression would be the W56R expression plasmid described in the 2020 Supekova paper, which has the Act1 promoter, single OXA1 or SU9 MTS, COX2ORF, in a high copy vector (i.e., moderate promoter, high copy, see fig 6 of 2020 PNAS paper). Was the OC-HC plasmid analyzed here (Fig 5) identical to the Supekova construct, or a construct made in this work that has similar elements? If the latter, then the comparison needs rewording.

The reference constructs OC-LC and OC-HC were assembled with our assembly strategy, replicating the respective constructs described by Supekova *et al.*. To clarify that these constructs were reconstructed based on the available information, we rephrased this section as follows:

“We then compared the respiratory growth benefit conferred by the optimised version (referred to as OV) to constructs expressing the COX2-W56R gene with one copy of OXA1 MTS under the control of the ACT1 promoter, replicating the functionally best expression system described by Supekova and collaborators (Supekova *et al.* 2010) (Figure 5.A). Respective reference constructs were made both with a high-copy (referred to as OC-HC) and a low-copy vector (referred to as OC-LC).”

1.19 *Line 597, line 614, and elsewhere: "massively" should be reworded.*

Throughout the manuscript, “massively” was replaced by “substantially”.

1.20 *Line 556-557: "Interestingly, we observed that several highly favourable mutations were concentrated in the second TMH" and*

Lines 571-573: "These results confirm that adjusting the TMH composition of allotopic transmembrane mitochondrial proteins is a key factor in a successful adaptation to nuclear relocation." and

Final sentence 627-630: "we propose that an experimental relocation of MT-CO2 in human cells should focus on engineering the amino acid sequences in the two transmembrane stretches to increase their $\mu\Delta G_{app}$, and particularly, testing substitutions at positions L73 and L75."

As discussed above, were these mutations shown to encode highly favorable variants when tested in isolation?" If not, then these statements should be altered..

In the revised manuscript, we have included testing of those most promising mutations individually. Please see our response to comment 1.1.

Reviewer 2

2.1 *Line 323: Do the authors really want to say pJEN1 or pICL1? The growth assay shown in Figure 2B would suggest the authors were trying to say ICL1. The growth assay on glycerol is intended to study fitness in respiratory conditions. However, it is hard to interpret, and it is hard to tell which one is growing better. So, I would recommend the authors repeat the assay and keep the plates for a longer time.*

We appreciate the reviewer for pointing this out. In line 323, we intended to convey that, compared to pACT1, the use of pADH1 and pJEN1 had a positive effect, whereas pICL1 did not provide any additional benefit. To emphasise the difference, we have repeated the assay with 7 days of incubation (see Figure 2).

2.2 *Growing yeasts on glycerol media will enhance lipid synthesis and thereby residual OCR from lipid metabolic pathways. So do the authors consider using a different fermentable carbon source for their assays?*

We agree with the reviewer about the potential influence of glycerol on residual OCR. Including the COX2 deletion strain in the OCR measurements helped account for this residual increase, ensuring that the differences observed between conditions are meaningful. Therefore, it did not appear essential to us to repeat the experiment using alternative carbon sources, such as ethanol or lactate.

Reviewer 3

3.1 *The authors should include an empty plasmid as a control in all spotting assays to demonstrate that the vector itself does not affect respiratory processes.*

We have included an “empty plasmid” control (Figure S1.C and Figure S6.B and C), which confirmed that the empty vector had no effect on respiratory growth.

3.2 *While the results are interesting and robust, the connection of this research to human mitochondrial disease and gene therapy seems tenuous.*

We thank the reviewer for highlighting this point. In order to strengthen the connection of this research to human mitochondrial disease, we performed a complementation assay trying to express a yeast codon-optimized version of the human *MT-CO2* gene (Figure S6.B and C). We tested the potential benefit of the Y40R, L73S and L75K mutations by introducing it in allotopic constructs devoid of the yeast *COX2* leader peptide sequence (Figure S6.B) or when this leader peptide was added (Figure S6.C). Unfortunately, we did not observe any restoration of the respiratory phenotype of the deletion mutant. However, this could be due to the absence of essential chaperones/assembly factors in yeast, which stresses the importance of mirroring the observed beneficial mutations in human cells rather than in yeast.

We have added the following sentences to present these results line 855-861: “ We explored the benefit of these mutations by allotopically expressing a yeast codon-optimised version of the human *MT-CO2* gene in the $\Delta\text{cox2}::\text{ARG8m}$ deletion mutant (Figure S6.B and S6.C). Unfortunately, none of the tested conditions lead to a significant restoration of the respiratory metabolism of the yeast mutant strain, probably due to the interplay of different chaperones and/or assembly factors between both organisms.(Watson and McStay 2020)”

3.3 *In the promoter screening experiment, it is unclear why the authors chose ICL1 for the OCR test and epPCR for sequence optimization, given that figure 2a suggests TDH3 could also be a good candidate.*

In the promoter screening experiment, our goal was to identify candidates with low expression levels under fermentation conditions and strong expression during respiratory conditions. Given that Cox2-W56Rp tends to aggregate, reducing its expression during fermentation helps both minimize cellular burden and lower the risk of inducing spontaneous adaptive mutations. The TDH3 promoter exhibited the highest expression during fermentation and only moderate expression in respiratory conditions. For the rationale of choosing pICL1 in particular, please also see comment 1.14.

3.4 *The statistical analysis of the OCR experiments shows significant differences between each mutant and the wild type. However, it is important to clarify if there are differences between the various promoters, high or low copy vectors, and different evolved coding sequences.*

We thank the reviewer for highlighting this point. To better represent the statistical differences between all the conditions, we have implemented a letter code. We have added the following sentence to the concerned figure legends: “A, B, C and D: Denotes a significant difference between the conditions ($p < 0.05$).”

3.5 *It is unclear why the high copy plasmid could already rescue the delta cox2::ARG8m effectively, yet the authors opted to optimize the low copy vector by changing promoters rather than directly optimizing the coding sequence in the high copy vector.*

As the reviewer pointed out, using a high-copy vector already significantly restored respiratory growth, which made it difficult to identify the specific benefits of the different optimizations. To address this, we opted for a low-copy vector to reduce cellular burden and achieve a better dynamic range for evaluating the advantages of the optimizations. After refining the expression system, we tested the impact of expressing the optimized version using a high-copy vector (Figure S3.D), but this did not provide any additional benefit.

3.6 *In figure 2a, it is interesting that most highly expressing promoters peak at 18 hours, whereas ethanol peaks at 41 hours. An explanation for this observation would be helpful.*

This is an interesting point. The variations in GFP fluorescence are likely linked to the growth rates of the different strains in either YPD or YPEtOH. For the former, we conducted the experiment in a standard 96-well plate, incubated for 24 hours in our plate reader, where OD600 nm and GFP signals were monitored every 10 minutes. However, we were unable to use the same setup for YPEtOH due to insufficient oxygenation in the media to support respiratory growth. Therefore, we cultured the strains in a 96-well deep-well plate, incubated at 30°C in a shaking incubator, and manually sampled aliquots from each well to measure OD600nm and GFP signal in a plate reader.

When transferred to YPEtOH, there was an initial lag in growth, corresponding to the metabolic shift from fermentation-based to respiration-based processes. Additionally, the growth rate reached in YPEtOH was lower compared to that in YPD. Both effects led to assay cultures in YPD reaching stationary phase much earlier than those in YPEtOH. Despite normalisation by OD600, these differences in growth dynamics (and linked GFP production, maturation and degradation dynamics) likely account for the differences in GFP fluorescence between the two media.

3.7 *What happens when the optimized final system is expressed in a high copy vector? Would it show even better performance?*

We tested this hypothesis and neither switching from a low- to high-copy vector nor adding a second copy of the OXA1 MTS were beneficial (see Figure S3.D).

3.8 *Including mRNA expression levels of strains expressing COX2 rescue vectors would strengthen the findings.*

While measuring the expression level of the allotopic COX2 gene could provide valuable insights, we believe that there is no direct correlation between mRNA abundance and the amount of mature Cox2p accumulated. As shown in Figure 1.D, the levels of mature Cox2p in the high-copy conditions are only twice as high as those in the low-copy conditions, despite the copy-number difference between high- and low-copy vectors being approximately 5- to 10-fold. This disparity is likely due to a significant amount of unprocessed Cox2p either aggregating or being degraded as a result of deficiencies in the import or processing of the premature protein. Therefore, we argue that monitoring the levels of mature and functional Cox2p and cellular oxygen consumption rates provide more relevant insight than mRNA expression.

3.9 *The U/M in the western blot figure should be clearly specified in the figure legend.*

The following was added to the concerned figure legends: "U: Unprocessed, M: Mature."

3.10 *There are some typos, please correct. For example: P21 line 442 at best, "thy"*

We hope to have caught all typos in subsequent through proof reads.

January 19, 2025

RE: GENETICS-2024-307723

Dear Dr. Cai:

I am pleased to accept your manuscript entitled "A new set of mutations in the second transmembrane helix of the Cox2p-W56R massively improves its alltopic expression in *Saccharomyces cerevisiae*." for publication in GENETICS, pending minor revision.

Please submit your revision along with a brief description of how you modified the manuscript in response to the reviewers' concerns and suggestions (which can be viewed at the bottom of this email. Most important are the comments from Reviewer #1, which requires careful text edits. Please revise the manuscript accordingly. Also please make sure that all data for graphs are made available as supplementary source files, per GENETICS requirements.. I expect you should be able to submit a revised manuscript within 30 days. A suitably revised manuscript will be acceptable for publication; I don't expect to send it out for review.

When revising the ms., please make an effort to shorten it, because that almost always improves a manuscript. We urge authors to heed the advice of Strunk and White: "omit needless words"¹. Follow this link to submit the revised manuscript: Link Not Available

Thank you for submitting this story to Genetics.

Sincerely,

Junbiao Dai
Associate Editor
GENETICS

Approved by:
Meera Sundaram
Senior Editor
GENETICS

Reviewer comments:

Reviewer #1 :

I attach two files, merged in single pdf. One with comments in response to the authors revisions. The second, a summary of the mutations and issues identified (as a guide to my review)

Reviewer #2 :

The manuscript is greatly improved due to the authors carefully addressing the concerns of each reviewer.

Reviewer #3 :

The reviewer has addressed most of the comments.

The author did not provide raw data for all the bar plots figure in the manuscripts. It would be better to make it available.

"western blot analysis confirmed the absence of Cox2p in our *cox2::ARG8m* deletion strain, and we saw a twice the amount of mature Cox2p in high-copy conditions compared to low-copy conditions (12% versus 6%, Figure 1.C)" This was talking about figure 1D?

Associate Editor comments:

Reviewer 1's evaluation of author's response to issues noted in reviewer 1's initial review:

Before assessing the revised manuscript, I found it necessary to create a summary of the various clones carrying missense variants identified in the screen, and the parental plasmids and COX2 gene sequences used as controls. In describing and presenting the results, the text of the manuscript is unclear and misleading in some places. This summary is attached. Given that the mutagenesis was performed starting with a parental plasmid carrying W56R plus two unintended mutations (I190V, and A21C) that were presumably introduced during the Cox2 commercial gene synthesis, I found it difficult to assess the experiments (controls, follow-up on individual missense mutations to assess functional consequence) without first creating the document attached for reference. The summary also shows that two missense mutations were not tested singly, but rather were discarded based on the locations of the affected amino acids in the 3-D structure of the protein.

The way I understand it is that there are three forms of the W56R Cox2 sequence relevant to these studies:

Benchmark:

1. Wt-W56R = Supekova allele (mitochondrial ORF DNA sequence carrying W56R)

This study:

2. Wt-W56R-(codon optimized to nuclear genetic code) = intended DNA synthesis product and starting point to mutagenesis
3. Wt-W56R-I190V-(A21C)-(codon optimized to nuclear genetic code) = parent to mutagenesis carrying two additional mutations

In reading the text and figures, it is at times difficult to know what particular W56R plasmid is being used as control to assess the effects of missense mutations. Plasmids #2 and #3 include a number of silent mutations relative to #1 intended as codon optimization for nuclear encoded expression. In theory, these mutations could affect mRNA translatability and/or mRNA dynamics (stability, transport, etc.). Plasmid #3 also carries the I190V missense mutation, therefore any identified missense mutations in the screen represent double missense mutants relative to plasmid #2 (or plasmid #1, for that matter).

As I understand it, the goal of this work is to improve upon the allotopic expression of the Wt-W56R Supekova allele (the benchmark to this study).

(As an aside, it would have been interesting to compare the Supekova allele to the codon optimized allele created here, using the same expression parameters, but this was not done. However, at least formally, any missense mutations that are attributed to further enhancement of W56R COX2 function in the mitochondria using a codon optimized version, would have to be compared to the codon optimized version, not the Supekova allele.)

1.1

"Importantly, the most beneficial amino acid substitutions found in the second transmembrane helix (L93S and I102K) are conserved residues in the corresponding positions of human COX2 (L73 and L75) and we propose that mirroring these changes could help

improve allotopic Cox2p expression in human cells." It was not clear to me that these specific residues were unambiguously shown to elicit the desired cis-effects, so this will have to be clarified if this statement is to stand.

Based on the issues highlighted in the attached summary, this concern has not been sufficiently addressed. As indicated in the summary and above, it is difficult to know (from the manuscript text) what control plasmids are being used to assess the effects of the individual or combined missense mutations. For example, is the control plasmid #2 or #3, or even #1 (see above)? The relevant control plasmid #3 appears to be missing in several of the analyses. Also, discarding analysis of the K3R, T10R missense mutations identified in the E8 and F7 double missense containing clones is unwarranted and makes this analysis incomplete, especially given that the specific missense mutations in E8 and F7 that were tested (prioritized based on their location in the 3D structure) were shown to have no effect.

In this regard, highlighting the selected missense mutations in figure 3E is misleading. The legend "(E) Annotation of the identified mutations present in the isolated epPCR clones E6, E8, C7, F7 and F9" is misleading. Highlighting non-functional mutations is not informative. Some of these are clearly passenger mutations and not relevant, and in addition, as noted, two identified missense mutations were not analyzed (and not shown), and could well be "functional". The I190V mutation was in the parental parent plasmid (ie, it was not identified in the selection screen).

As noted in the summary, the rationale for combining mutations that have no effect singly is unclear, especially as not all mutations were tested singly. The (modest) effect of the L93S I102K double mutant does not appear significantly different that the E6 plasmid carrying I195K.

Has the effect of the I190V mutation, present in the parental plasmid to the mutagenesis, been considered?

1.2

The authors respond that the small fraction containing the unintended mutations were unexpectedly enriched during the error prone PCR, explaining their presence in the isolated clones. This seems highly unlikely (selective PCR, or selection in E.coli during library prep). More likely that a subclone was expanded to prep the parental plasmid used as template for the PCR, and it contained the unintended mutations (I190V and A21C) clonally.

One concern is that the authors state that these mutations were present in a small fraction of the PCR template. This raises the concern as to whether this template plasmid prep was used as the control in validation experiments. If so, then control plasmid transformants will lack the I190V missense mutation common to all the isolates, and therefore not be a proper control.

1.5

Regarding the combined effects L93S I102K, see above.

1.8

Validation of primary isolates should have been done by isolation of the plasmid, retransformation, and testing replicate transformants. Performing growth on the primary isolates does not validate the plasmid that is isolated and sequenced. The fact that the C7 and F9 differences in the phenobooth retesting of primary isolates are considered spurious highlights this point.

1.9

I still disagree that the TMH2 mutations can be interpreted as representing a ‘hotspot’.

Lines 393-419: I disagree that a hotspot was identified in TMH2 (line 393-395). “To investigate the role of the identified mutations...”(line 400-405); only a subset were tested (see above). “as TMH2 mutations substantially improve respiratory growth....” (line 406): the individual mutations tested did not improve growth.

1.18

I believe that the W56R Supekova “original construct” tested in this work is significantly different than the W56R construct in the Supekova paper (ie, does not necessarily replicate it). For example, the W56R ORF has been codon optimized. It is possible, for example, that these RNA sequence changes reduce benefit. Were the unintended I190V and A21C mutations present? If so, these could reduce benefit.

1.20

I still believe these statements should be altered based on the data.

Summary of mutations in Cox2 and related issues

Wt

Wt-W56R = Supekova allele (mitochondrial ORF DNA sequence carrying W56R)

Wt-W56R-I190V-(A21C)-(codon optimized to nuclear genetic code) = parent to mutagenesis

I190V not “selected”, rather a pre-existing missense mutation in parental plasmid

-PARENTAL PLASMID (apparently contains 2 codon DNA synthesis errors?):

Wt-W56R +I190V +A21C (silent)

Clones picked and sequenced:

Additional mutations in isolated clones:

E6	(W56R, I190V , A21C)	I95K ,	0 more silent
E8	(W56R, I190V , A21C)	I102K , K3R	3 more silent
F7	(W56R, I190V , A21C)	T10R , L210M	0 more silent
C7	(W56R, I190V , A21C)	L93S	0 more silent
F9	sib of C7		

Testing effects of individual mutations:

Figure S1:

“Spot test presenting the benefit conferred by the different mutations when introduced individually in the COX2-W56R gene”

--Tested singles in context of Wt-W56R

Issues:

Why test singles in the context of Wt-W56R?

What about I190V context? Codon optimization differences? A21C? in parental plasmid.

Why only 5 mutations out of 7?

E8: Why only test I102K? (it had no effect) **K3R not tested individually (fig S1)**

F7: Why only test L210M? (it had no effect) **T10R not tested individually (fig S1)**

Mutations tested singly: I190V (parental missense), L210M, I95K, I102K, L93S

I190V single no effect

I102K single no effect, **K3R not tested?** (E8 double missense)

L210M single no effect, **T10R not tested?** (F7 double missense)

L93S single no effect

I195K single modest effect

Missing; K3R, T10R

Missing control parental plasmid (W56R-I190V-(A21C)-codon optimized

Single isolates tested, no replicates?

Figure 3A: **Missing control parental plasmid (W56R-I190V-(A21C)-codon optimized**

Figure 3B: I95K, I102K, L93S (3 pairwise combos and triple)
Rationale unclear (combining mutations that have no effect)
Also, K3R, T10R were untested as singles
Were these tested in context of I190V, parental plasmid sequence?
Where is parental plasmid control (W56R-I190V-(A21C)-codon optimized

Figure 3C: Why only 6 mutations shown on structure (W56R + I190V + 4/6 missenses identified)?
Missing; K3R, T10R

Figure 3 legend:

“(E) Annotation of the identified mutations present in the isolated epPCR clones E6, E8, C7, F7 and F9.”

1. The figure does not show all identified mutations. The two mutations not tested as singles may well be functionally relevant.
2. The figure shows codon changes that were shown to have no effect; inclusion is therefore misleading. Codons with no effect are passenger mutations and are not relevant.
3. I190V was not identified by the growth selection of the screen- it was an unintended pre-existing missense mutation in the parental plasmid.

Reviewer 1

1.1. *"Importantly, the most beneficial amino acid substitutions found in the second transmembrane helix (L93S and I102K) are conserved residues in the corresponding positions of human COX2 (L73 and L75) and we propose that mirroring these changes could help improve allotopic Cox2p expression in human cells." It was not clear to me that these specific residues were unambiguously shown to elicit the desired cis-effects, so this will have to be clarified if this statement is to stand.*

Based on the issues highlighted in the attached summary, this concern has not been sufficiently addressed. As indicated in the summary and above, it is difficult to know (from the manuscript text) what control plasmids are being used to assess the effects of the individual or combined missense mutations. For example, is the control plasmid #2 or #3, or even #1 (see above)? The relevant control plasmid #3 appears to be missing in several of the analyses. Also, discarding analysis of the K3R, T10R missense mutations identified in the E8 and F7 double missense containing clones is unwarranted and makes this analysis incomplete, especially given that the specific missense mutations in E8 and F7 that were tested (prioritized based on their location in the 3D structure) were shown to have no effect.

It is well known that a major constraint limiting an efficient nuclear relocation of mitochondrial genes is the high hydrophobicity of their encoded proteins, which are prone to aggregating in the cytoplasm and/or not being properly imported and sorted. This was well demonstrated by the benefit brought by the W56R mutation previously presented by Supekova et al. This mutation resulted in an important decrease in hydrophobicity and was mandatory to allow partial restoration of respiratory growth.

In the present paper, we not only aimed to discover potential additional functional mutations that would improve the artificial nuclear relocation of Cox2p in yeast, but also to shed light on potential conserved residues (between yeast and human) which upon substitution could provide a benefit in both species. As benchmark/ baseline to improve upon, Cox2-W56R (codon optimised to nuclear genetic code) was used throughout the work.

As for the substitutions L93S and I102K in the second transmembrane helix, both provide a respiratory improvement individually (Figure S1), and we found their combination to result in a pronounced improvement. We have performed comprehensive functional and biochemical characterisations of the respective construct.

We also tested whether using the MTS sequence from clone F7 would be beneficial (Figure S3.A) but did not observe any improved benefit. So, we decided to focus on the mutations identified in TMH2, which unlike mutations in the MTS can be modelled to the human gene. We hope this explains the rationale of omitting K3R and T10R from further analysis, as raised by this reviewer.

In this regard, highlighting the selected missense mutations in figure 3E is misleading. The legend "(E) Annotation of the identified mutations present in the isolated epPCR clones E6, E8,

C7, F7 and F9" is misleading. Highlighting non-functional mutations is not informative. Some of these are clearly passenger mutations and not relevant, and in addition, as noted, two identified missense mutations were not analyzed (and not shown), and could well be "functional". The I190V mutation was in the parental parent plasmid (ie, it was not identified in the selection screen).

We agree with the reviewer and have modified the figure accordingly removing the passenger/non-functional mutations. Again, regarding the mutations identified in MTS', they could not be transposed to human so we decided to focus on those present in the protein sequence. Finally, we have modified the Figure title accordingly: "(E) Annotation of the functional mutations identified in the protein sequence in the isolated epPCR clones E6, E8, C7 and F9"

As noted in the summary, the rationale for combining mutations that have no effect singly is unclear, especially as not all mutations were tested singly. The (modest) effect of the L93S I102K double mutant does not appear significantly different that the E6 plasmid carrying I195K.

We explored the benefit of these different mutations, independently, and reported the results in Figure S1.C. After 4 days, both L93S and I102K appeared to confer a minor benefit, albeit a much less pronounced one than that provided by I95K. We tested different combinations of these three mutations for further improving the respiratory phenotype. Indeed, the combination of L93S and I102K proved to be the most functional one, outperforming combinations with I95K, which individually is the most beneficial of the sampled mutations. This goes to show that combinatorially exploring sampled mutations can provide non-obvious optima.

Has the effect of the I190V mutation, present in the parental plasmid to the mutagenesis, been considered?

As presented in Figure S1.C, we tested the benefit conferred by the mutation I190V alone and did not observe any benefit.

1.2. *The authors respond that the small fraction containing the unintended mutations were unexpectedly enriched during the error prone PCR, explaining their presence in the isolated clones. This seems highly unlikely (selective PCR, or selection in E.coli during library prep). More likely that a subclone was expanded to prep the parental plasmid used as template for the PCR, and it contained the unintended mutations (I190V and A21C) clonally.*

The plasmid extract used for random mutagenesis has been Sanger sequenced and the sequencing traces are presented below:

The upper sequencing trace picture reveals, in the plasmid extract, a dual population with a mixture of A and C at position 21 (R7). Similarly, the lower sequencing trace picture reveals a dual population with a mixture of A and G at position 709 (I190). These results confirm the presence of a fraction of the plasmids containing the unintended mutations and rules out the possibility to have selected a bacterial clone containing these two mutations.

One concern is that the authors state that these mutations were present in a small fraction of the PCR template. This raises the concern as to whether this template plasmid prep was used as the control in validation experiments. If so, then control plasmid transformants will lack the I190V missense mutation common to all the isolates, and therefore not be a proper control.

This plasmid extract was transformed in our deletion strain and used as the control in all the validation experiments. As demonstrated, the I190V mutation had no effect at all. For each experiment, we started from multiple fresh individual yeast clones and never observed any difference in the growth pattern of the deletion mutant strain + control plasmid across the experiment presented in the paper. Neither the silent mutation A21C nor the missense mutation A709G had any effect on the respiratory growth of the deletion mutant.

1.5. *Regarding the combined effects L93S I102K, see above.*

Regarding our reply to the reviewer's comment, please see above.

1.8. *Validation of primary isolates should have been done by isolation of the plasmid, retransformation, and testing replicate transformants. Performing growth on the primary isolates does not validate the plasmid that is isolated and sequenced. The fact that the C7 and F9 differences in the phenobooth retesting of primary isolates are considered spurious highlights this point.*

We did analyse, in the first place, the respiratory phenotype of the nine isolated better growing clones on the primary isolates. After confirmation, as described in the material section, we extracted the plasmids, sequenced them, and re-transformed them in the deletion strain, along with the plasmids containing the different combinations of L93S, I95K and I102K mutations (Figure 3.B). The results presented in Figure 3.B, after retransforming the plasmids extracted from epPCR clones E6, E8 and C7 yielded the same phenotype than that observed on the primary isolates.

Regarding the differences observed with the PhenoBooth, this is a technical artefact associated with the assay, which is for instance liable to edge- and other positional effects. Further, it samples colonies from one plate to another, and thus, inoculum masses can vary. This is particularly highlighted by the results presenting the size of the colonies formed on YPGly (Figure S1.B). While a few constructs formed larger colonies than the wild type in the screening assay for respiratory growth phenotype (Figure 3.A), none of these colonies actually grew better than the wild type. This is an actual artefact that we are aware of, but despite this, it enabled to successfully isolate better growing clones, which was the primary purpose of this experiment.

1.9. *I still disagree that the TMH2 mutations can be interpreted as representing a 'hotspot'. Lines 393-419: I disagree that a hotspot was identified in TMH2 (line 393-395). "To investigate the role of the identified mutations..."(line 400-405); only a subset were tested (see above). "as TMH2 mutations substantially improve respiratory growth..." (line 406): the individual mutations tested did not improve growth.*

The different instances of "hotspot" were replaced in the manuscript:

Line 174-175: “Mutational hotspots were explored further by combining identified mutations using the epPCR clone E6 vector” was replaced by “**The additive effect of mutations identified in TMH2 was explored further by combining identified mutations using the epPCR clone E6 vector**”

Line 388-389: “**In addition to these mutations**, a hotspot of beneficial mutations has been identified in the TMH2, with three substitutions” was replaced by “**In addition to these mutations, multiple beneficial mutations have been identified in the TMH2, with three substitutions**”

Line 401: “As the TMH2 hotspot mutations **substantially** improved respiratory growth” we removed the word hotspot.

1.18. *I believe that the W56R Supekova “original construct” tested in this work is significantly different than the W56R construct in the Supekova paper (ie, does not necessarily replicate it). For example, the W56R ORF has been codon optimized. It is possible, for example, that these RNA sequence changes reduce benefit. Were the unintended I190V and A21C mutations present? If so, these could reduce benefit.*

We do agree that the construct used in this study did not always perfectly mirror the results obtained with the constructs used in Supekova paper. However, the results presented in Figure 1.B seem to reflect the results presented in Supekova paper Figure 6. Particularly, in both cases, expressing the 1xOXA1/SU9 MTS-COX2-W56R from a multicopy vector under the control of ACT1 promoter significantly restored the respiratory growth of the deletion mutant. Even if we obtained different results with 2 copies of these MTS, throughout the paper, we only focused on using 1 copy of the different MTS. Again, as presented in Figure 5.A, the results obtained expressing 1xOXA1 MTS-COX2-W56R multicopy vector under the control of ACT1 promoter, which we referred to as the original expression system described in Supekova paper, are highly similar to those presented in Supekova paper’s Figure 6

We did observe some differences in the level of accumulated mature Cox2p as compared to those presented in the Supekova paper, but rather than attributing these differences to nuclear codon optimisation, we believe they predominantly stem from intentional differences in the experimental setup. In the present manuscript, all the precultures were performed in fermentable carbon source-containing media and were grown without agitation. Unreported observations indicated that growing precultures of cells expressing the 1xOXA1 MTS-COX2-W56R under the control of ACT1 promoter from a multicopy vector with agitation was resulting in the accumulation of spontaneous mutants carrying favourable nuclear mutations improving the respiratory phenotype. Since this would impair our study, we decided to grow the precultures without agitation which might explain the lower accumulation of mature Cox2p and some differences between the system described herein and that from Supekova and collaborators.

1.20. *I still believe these statements should be altered based on the data.*

The initial statements were:

Line 556-557: "Interestingly, we observed that several highly favourable mutations were concentrated in the second TMH" and

Lines 571-573: "These results confirm that adjusting the TMH composition of allotopic transmembrane mitochondrial proteins is a key factor in a successful adaptation to nuclear relocation." and

Final sentence 627-630: "we propose that an experimental relocation of MT-CO2 in human cells should focus on engineering the amino acid sequences in the two transmembrane stretches to increase their $\mu\Delta G_{app}$, and particularly, testing substitutions at positions L73 and L75."

As discussed above, were these mutations shown to encode highly favorable variants when tested in isolation?" If not, then these statements should be altered..

In the revised manuscript, we have tested the benefit of these mutations individually (Figure S1.C). We did observe that individually, these mutations were bringing a benefit, which was the most noticeable for the mutation I95K. We maintain that the modulation of TMH hydrophobicity is essential and that the role of these mutations in human should be explored.

However, we do agree with the reviewer regarding mitigating the statement on the benefit of each mutation and modified the manuscript as follow:

"Interestingly, we observed that several favourable mutations were concentrated in the second TMH"

Reviewer 3

3.1 *The author did not provide raw data for all the bar plots figure in the manuscripts. It would be better to make it available.*

We have provided the raw data in Table S4.

3.2 *"western blot analysis confirmed the absence of Cox2p in our cox2::ARG8m deletion strain, and we saw a twice the amount of mature Cox2p in high-copy conditions compared to low-copy conditions (12% versus 6%, Figure 1.C)" This was talking about figure 1D?*

We agree with the reviewer and modified the manuscript as follow:

"Western blot analysis confirmed the absence of Cox2p in our cox2::ARG8m deletion strain, and we saw a twice higher amount of mature Cox2p in high-copy conditions compared to low-copy conditions (12% versus 6%, Figure 1.D)"

February 23, 2025

RE: GENETICS-2024-307723R1

Prof. Yizhi Cai
The University of Manchester
Manchester Institute of Biotechnology
131 Prince Street
Manchester, N/A M1 7DN
United Kingdom

Dear Dr. Cai:

Congratulations! We are delighted to inform you that your manuscript titled "A new set of mutations in the second transmembrane helix of the Cox2p-W56R massively improves its allotopic expression in *Saccharomyces cerevisiae*." is acceptable for publication in GENETICS. Many thanks for submitting your research to the journal.

To Proceed to Production:

1. Format your article according to GENETICS style, as discussed at <https://academic.oup.com/genetics/pages/general-instructions>, and upload your final files at <https://genetics.msubmit.net>.
2. Your manuscript will be published as-is (unedited-as submitted, reviewed, and accepted) at the GENETICS website as an Advanced Access article and deposited into PubMed shortly after receipt of source files and the completed license to publish. Please notify sourcefiles@thegsajournals.org if you do not wish to publish your article via Advanced Access.
3. We invite you to submit an original color figure related to your paper for consideration as cover art. Please email your submission to the editorial office or upload it with your final files. You can submit a small-sized image for evaluation, and if selected, the final image must be a TIFF file 2513px wide by 3263px high (8.375 by 10.875 inches; resolution of 600ppi). Please avoid graphs and small type.

If you have any questions or encounter any problems while uploading your accepted manuscript files, please email the editorial office at sourcefiles@thegsajournals.org.

Sincerely,

Junbiao Dai
Associate Editor
GENETICS

Approved by:
Meera Sundaram
Senior Editor
GENETICS

note: Please add jnls.author.support@oup.com and genetics.oup@kwglobal.com (or the domains @oup.com and @kwglobal.com) to your email program's "safe senders" list. You will be contacted by both at various points during the production process.

Review comments (if applicable):